# CodeTransOcean: A Comprehensive Multilingual Benchmark for Code Translation

**Weixiang Yan**[1*]    **Yuchen Tian**[2*]    **Yunzhe Li**[3]    **Qian Chen**[4]    **Wen Wang**[4]

[1]University of California, Santa Barbara   [2]The University of Hong Kong
[3]University of Illinois at Urbana-Champaign   [4]Speech Lab, Alibaba Group
weixiangyan@ucsb.edu   yuchent@connect.hku.hk
yunzhel2@illinois.edu   {tanqing.cq,w.wang}@alibaba-inc.com

## Abstract

Recent code translation techniques exploit neural machine translation models to translate source code from one programming language to another to satisfy production compatibility or to improve efficiency of codebase maintenance. Most existing code translation datasets only focus on a single pair of popular programming languages. To advance research on code translation and meet diverse requirements of real-world applications, we construct **CodeTransOcean**, a large-scale comprehensive benchmark that supports the largest variety of programming languages for code translation. CodeTransOcean consists of three novel multilingual datasets, namely, **MultilingualTrans** supporting translations between multiple popular programming languages, **NicheTrans** for translating between niche programming languages and popular ones, and **LLMTrans** for evaluating executability of translated code by large language models (LLMs). CodeTransOcean also includes a novel cross-framework dataset, **DLTrans**, for translating deep learning code across different frameworks. We develop multilingual modeling approaches for code translation and demonstrate their great potential in improving the translation quality of both low-resource and high-resource language pairs and boosting the training efficiency. We also propose a novel evaluation metric *Debugging Success Rate@K* for program-level code translation. Last but not least, we evaluate LLM ChatGPT on our datasets and investigate its potential for fuzzy execution predictions. We build baselines for CodeTransOcean and analyze challenges of code translation for guiding future research. The CodeTransOcean datasets and code are publicly available at https://github.com/WeixiangYAN/CodeTransOcean.

---

[*]Equal contribution. Work is supported by Speech Lab, Alibaba Group.

## 1 Introduction

Early software systems are developed using programming languages such as Fortran and COBOL, which have a significantly smaller user base compared to modern mainstream programming languages (e.g., Python and Java). Hence maintaining and modernizing early software systems are expensive (Opidi, 2020). Moreover, the readability and compatibility of the mixed multitude of programming languages are challenging when migrating existing software systems to new technology ecosystems or integrating software systems using different programming languages. The code translation task aims to convert source code from one programming language to another and is of great value in industry.

Code translation methods evolve from the inefficient, costly, and error-prone manual rewriting method to automatic methods. Automatic code translation methods can be categorized into *compilers and transpilers*, *rule-based methods*, and *neural network based methods*. Neural models (Feng et al., 2020; Wang et al., 2021, 2023b) have become dominant in code translation. Details of code translation methods are presented in Appendix A.1. The performance of neural models relies heavily on large-scale high-quality parallel data. However, existing code translation datasets are limited by **insufficient coverage of programming languages and mostly focusing on a single pair of popular programming languages**, **limited scale**, and **uneven data distribution**. The widely used CodeTrans (Lu et al., 2021) is a small dataset containing only Java-C# parallel data for quite short code samples. Other datasets (Ahmad et al., 2023; Rozière et al., 2020; Zhu et al., 2022b; Nguyen et al., 2013; Chen et al., 2018) suffer from the same limitations. Consequently, existing code translation models (Feng et al., 2020; Wang et al., 2021; Ahmad et al., 2021) are confined to a narrow range of one-to-one code

| Category | Language/Framework | Dataset Name | Train/Dev/Test #Samples | Avg. #Tokens/Sample | Avg. Length |
|---|---|---|---|---|---|
| Multilingual | Python, C, C++, Visual Basic, Go, PHP, Java, C# | MultilingualTrans | 19,115 / 3,759 / 7,545 | 398 / 421 / 491 | 1099 / 1135 / 1358 |
| | Swift, R, Rust, Fortran, Ada, Perl, COBOL, Lua, ... | NicheTrans | 165,457 / 23,509 / 47,502 | 292 / 375 / 505 | 785 / 995 / 1372 |
| | Python, C, C++, Visual Basic, Go, PHP, Java, C# | LLMTrans | – / – / 350 | – / – / 270 | – / – / 745 |
| Cross-Framework | PyTorch, TensorFlow, MXNet, Paddle | DLTrans | 282 / 36 / 90 | 625 / 1102 / 875 | 1318 / 2441 / 1841 |

Table 1: Summary of our **CodeTransOcean**. We report #Samples, Avg. #Tokens/Sample and Avg. Length for Train/Dev/Test sets of each dataset. Note that LLMTrans is only for testing. #Samples are on the **program-level**. #Tokens are based on RoBERTa tokenizer (Liu et al., 2019). Length is the number of characters.

translation scenarios. Moreover, deep learning has been broadly used and achieved unprecedented success. However, there are barriers between different deep learning frameworks during the actual production process. Existing code translation datasets also neglect important demands from real-world applications, including **modernizing early software systems developed in *niche* programming languages** and **migrating code across different deep learning frameworks**.

To address these limitations and advance neural code translation models, we construct a large-scale comprehensive multilingual code translation benchmark **CodeTransOcean**, summarized in Table 1. CodeTransOcean is an innovative benchmark that aims to provide a **unified platform** for evaluating various models *on a comprehensive set of code translation tasks* that reflect real-world demands. Based on this goal, each dataset in Code-TransOcean is specifically designed to tackle a key challenge in the field of code translation. Code-TransOcean includes three *multilingual* datasets, namely, the **MultilingualTrans** dataset (including eight popular programming languages), the **NicheTrans** dataset (translating between thirty-seven niche programming languages and the eight popular ones[1]), and a specialized dataset **LLMTrans** (including 350 data samples and their executed results) to evaluate executability of code translated by large language models (LLMs), and a *cross-framework* dataset **DLTrans** facilitating our proposed task for translating code between deep learning frameworks to enhance code reusability.

DLTrans includes 408 samples covering four mainstream deep learning frameworks.

Multilingual modeling shows great potential in neural machine translation (Aharoni et al., 2019; Wang et al., 2020; Zhu et al., 2023), but it has not been systematically explored for code translation. We investigate multilingual modeling for code translation using our MultilingualTrans, NicheTrans, and DLTrans datasets. Experimental results demonstrate that multilingual modeling significantly improves translation quality for both *high-resource* and *low-resource* language pairs and improves the model training efficiency.

Recent research indicates that the proficiency of the LLM ChatGPT in natural language translation is on par with commercial-grade translation systems (Jiao et al., 2023). **To the best of our knowledge, our work is the first to systematically investigate the potential of ChatGPT in code translation**. We develop a fully automated translation-execution-evaluation pipeline **AutoTransExecuter** to support this study. Note that *match-based metrics* and *execution-based metrics* have been used for evaluating code translation methods, with details in Appendix A.1. In order to accurately evaluate the usability of translated code from ChatGPT, we propose a novel execution-based evaluation metric **Debugging Success Rate @K (DSR@K)**, which is the percentage of samples with translation results that successfully execute and produce the expected functionality after $K$ debugging rounds. On our LLMTrans dataset, the baseline ChatGPT setting achieves 48.57% DSR@0. We find that self-debugging and one-shot improve the performance while chain-of-thought strategies degrade the translation accuracy. Since our AutoTransEx-

---

[1] We define popular and niche programming languages based on the TIOBE Programming Community Index, which is a metric of the popularity of programming languages.

ecuter still cannot cover arbitrary programming languages, we also propose a novel metric *fuzzy execution*, attempting to address the limitations of existing evaluation metrics for code translation. Our preliminary study using ChatGPT shows that ChatGPT is still inadequate to predict fuzzy execution for any arbitrary programming language, which demands future research.

Our contributions can be summarized as follows:

- **A large-scale multilingual code translation benchmark**: CodeTransOcean covers the largest number of popular and niche programming languages so far with the largest scale. It also includes an unprecedented dataset for translating code across different deep learning frameworks and a dataset and an automated pipeline for evaluating LLMs on code translation. We establish baselines for all datasets in CodeTransOcean.
- **Multilingual modeling for code translation**: We are the first to systematically evaluate multilingual modeling on code translation for both high-resource and low-resource language pairs. Experimental results demonstrate that multilingual modeling significantly improves translation quality for both *high-resource* and *low-resource* language pairs and improves training efficiency.
- **ChatGPT on code translation**: We conduct the first comprehensive study of the potential of ChatGPT on code translation, investigating efficacy of prompting strategies, hyperparameters, self-debugging, One-shot, and Chain-of-Thought.
- **New evaluation metrics**: We propose *DSR@K* to evaluate translation and debugging capabilities of LLMs. We also propose a *fuzzy execution* metric based on LLMs and conduct a preliminary study using ChatGPT on this metric.

## 2 Related Work

**Code Translation Datasets** The success of neural models for code translation relies heavily on large-scale high-quality parallel data. However, existing code translation datasets are plagued by issues such as *insufficient coverage of programming languages*, *limited scale*, and *imbalanced data distribution*. The widely used code translation dataset CodeTrans (Lu et al., 2021) in the CodeXGLUE benchmark consists of Java-C# function pairs. The small parallel corpus AVATAR (Ahmad et al., 2023) is constructed for Java-Python code translation. Nguyen et al. (2013) construct a Java-C# dataset to explore statistical machine

translation on code translation tasks[2]. Chen et al. (2018) explore this dataset from Nguyen et al. and also construct a CoffeeScript-JavaScript parallel dataset for investigating tree-to-tree neural models for code translation. Rozière et al. (2020) create a dataset containing 852 programs to evaluate unsupervised methods. Recently, Zhu et al. (2022b) construct a new translation dataset CoST from the GeeksForGeeks website[3]. Subsequently, they release the translation dataset XLCoST (Zhu et al., 2022a), which expands the CoST dataset by 7.3 times. However, the limited language coverage of these datasets and their imbalanced data distribution hinder their practical applications. Rozière et al. (2022) construct the TransCoder-ST dataset to perform unsupervised code translation using automated unit tests. Details of these datasets are summarized in Table 2. Rithy et al. (2022) proposes a code translation dataset XTest containing nine programming languages with unit tests, but it is not open-sourced[4]. Although CodeNet (Puri et al., 2021) comprises many problem statements and provides corresponding solutions, experts have proven that about half of the CodeNet dataset has incorrect solutions (Zhu et al., 2022b), making it unsuitable for code translation tasks. With the limitations of existing code translation datasets, neural models trained on them may encounter overfitting, underfitting, and poor generalizability. Clearly, these issues impede the development of neural models for code translation. Therefore, constructing datasets that effectively address these problems is critical to enhance performance of code translation algorithms.

**Code Translation Methods and Evaluation Metrics** Details of code translation methods and evaluation metrics are presented in Appendix A.1.

## 3 The CodeTransOcean Benchmark

In this section, we provide detailed descriptions and analyses of our CodeTransOcean benchmark, including the code translation tasks, their associated datasets, and dataset statistics. Details of data collection methods and licensing information as well as quality control and quality assessment are presented in Appendix A.2. Note that the vast

---

[2]It was not possible to count specific information about this dataset because it was not released to the public and we were unable to obtain response from the authors.

[3]In Table 2, we report the **program-level** counts for the CoST dataset to facilitate a fair comparison with our own program-level datasets.

[4]We tried to contact the authors but there was no response.

| Dataset Source | Programming Languages | #Samples | Avg. #Tokens/Sample | Avg. Length |
|---|---|---|---|---|
| CodeTrans (Lu et al., 2021) | Java, C# | 11,800 | 59 / 63 / 58 | 205 / 218 / 202 |
| Avatar (Ahmad et al., 2023) | Java, Python | 9,517 | 239 / 235 / 234 | 691 / 687 / 688 |
| Nguyen et al.(Nguyen et al., 2013) | Java, C# | 16,966 | – | – |
| Lachaux et al.(Rozière et al., 2020) | C++, Java, Python | 852 | - / 119 / 120 | - / 313 / 311 |
| CoST (Zhu et al., 2022b) | C++, Java, Python, C#, Javascript, PHP, C | 16,738 | 272 / 180 / 199 | 770 / 458 / 511 |
| TransCoder-ST (Rozière et al., 2022) | Java, C++, Python | 437,030 | – | – |
| XLCoST (Zhu et al., 2022a) | C++, Java, Python, C#, Javascript, PHP, C | 122,151 | 234 / 232 / 222 | 644 / 634 / 606 |

Table 2: Summary of existing code translation datasets. For #Samples, we report **program-level** counts for Avatar, CoST, and XLCoST. Given that the original samples from other datasets are not organized at the program-level, we report counts at the snippet-level for these datasets. Avg. #Tokens/Sample and Avg. Length are counted in the same way as Table 1.

majority of the samples in CodeTransOcean provides explicit input and output, which is equivalent to unit tests. Overall, CodeTransOcean consists of 270,507 samples (over 200K unit tests), covering 45 programming languages for multilingual code translation and 4 deep learning frameworks for cross-framework code translation[5]. Note that all samples in all CodeTransOcean datasets are constructed at the **program-level**. We ensure a balanced distribution of each language/framework when constructing the datasets (Appendix A.2). There is no overlap between CodeTransOcean datasets and existing code translation datasets.

## 3.1 Multilingual Code Translation

With the increasing need to unify the language variety when implementing system integration or extensions with multilingual programming environments, we construct the MultilingualTrans dataset for multiple popular programming languages[6]. Among programming languages in the rankings, we select the Top-10 languages as popular ones except JavaScript and SQL[7] and construct the MultilingualTrans dataset based on the 8 pro-

gramming languages. We treat the other languages in the rankings as niche languages and construct the NicheTrans dataset for translating between niche languages and popular languages. Additionally, in order to quantitatively evaluate the execution capabilities of the code generated by LLMs (e.g., ChatGPT, PaLM2 (Anil et al., 2023)), we construct LLMTrans, which includes the execution results for a subset of MultilingualTrans and facilitates evaluating LLMs for multilingual code translation.

**MultilingualTrans Dataset** This dataset contains 30,419 program samples covering eight popular programming languages, namely, C, C++, C#, Java, Python, Go, PHP, and Visual Basic. Table 11 shows the statistics of each language pair. Note that XLCoST (Zhu et al., 2022a) is the only existing multilingual code translation dataset. Compared to XLCoST, MultilingualTrans is advantageous in more balanced data distribution across various programming languages, practicality of language pairs, and data quality. For example, the real-world requirement for translating Java into JavaScript as in XLCoST is quite limited. As to data quality, our MultilingualTrans originates from a programming chrestomathy website, with all data already reviewed and verified by the website.

**NicheTrans Dataset** The NicheTrans dataset contains 236,468 program samples, covering code translation pairs from thirty-seven niche programming languages, including Ada, COBOL, Pascal, Perl, Erlang, Fortran, Scala, Julia and others, to the eight popular ones. Table 12 shows statistics of each niche language. Although many studies have highlighted the practical necessity of code translation for modernizing niche programming languages (Chen et al., 2018; Zhu et al., 2022b; Rozière et al.,

---

[5]Code Translation also extends to conversions between different versions of the same language, e.g., Python 2 to Python 3. However, according to our survey, these translation tasks are quite straightforward. Naive Copy methods, specific translation tools, and tutorials (e.g., Python 2 to 3 Conversion Guide) already achieve high translation accuracy. As a result, we no longer include these types of tasks in our benchmark.

[6]We categorize languages as popular or niche based on the TIOBE Index Programming Language Rankings released in April 2023 https://www.tiobe.com/tiobe-index/.

[7]It is important to note that JavaScript and SQL, both within the top 10, are mainly used for front-end programming and database management respectively, signifying considerable differences in their usage scenarios compared to the other 8 languages.

2020), our NicheTrans dataset is the first dataset for code translation between these niche languages and popular ones. We believe this dataset will not only facilitate modernization of outdated programming languages more effectively, but also augment and evaluate generalizability of neural models.

**LLMTrans Dataset** The LLMTrans dataset aims to provide a benchmark for evaluating the performance of LLMs on code translation. The dataset translates seven popular programming languages to Python, totaling 350 program samples. We compile and test these samples and record the execution results. Based on this dataset, we design and implement an automated pipeline, **AutoTransExecuter**[8], automatically using LLMs to conduct code translation, execution, debugging, and calculating the success rate. This dataset and the automated pipeline ease investigation of the actual debugging success rate of LLMs on code translation and effectively measure the practical usability of LLMs. Details of the LLMTrans dataset are in Table 1.

## 3.2 Cross-framework Code Translation

**Cross-Deep-Learning-Framework Translation Task** The widespread applications of deep learning (DL) has spawned emergence of various DL frameworks, such as PyTorch, TensorFlow, MXNet, and Paddle. However, there are significant differences in syntax and dependency libraries between different frameworks, severely impeding reusability of projects[9]. Moreover, studies illustrate significant disparities in energy consumption and economic costs during training and inference between various frameworks (Georgiou et al., 2022). Selecting an appropriate DL framework for green AI has become paramount in an era of large models (Ananthaswamy, 2023). Code reusability and energy-economic efficiency in DL have emerged as critical considerations for both research and practical engineering implementation. Converting code between different DL frameworks is challenging, mainly due to differences between frameworks, code complexity, structural inconsistencies, and cross-platform compatibility (more details are in Appendix A.3). Existing cross-DL-framework adaptive technologies such as the ONNX[10] model conversion protocol require both parties to import

and export based on agreed data formats or to convert only the final model through the computation graphs. These technologies have obvious limitations. In contrast, we propose a **Cross-DL-framework Translation** task for code migration between different DL frameworks through code translation (Appendix A.4). Compared to existing cross-framework adaptive technologies, Cross-DL-framework Translation achieves re-implementation under multiple DL frameworks through an automated process, which not only generates highly readable code and enables secondary development, but also provides developers with flexibility on combining advantages of multiple frameworks.

**DLTrans Dataset** We construct the **DLTrans dataset** for Cross-DL-framework Translation, including four deep learning frameworks and spanning twelve directions. To the best of our knowledge, our work is the first to define the cross-DL-framework translation task and construct a corresponding dataset. We create two subsets of different granularities based on the collected code, namely, *coarse-grained* at the program level and *fine-grained* at the function or class level. Each code pair comprises code that shares the same functionality but is written in different popular DL frameworks, including PyTorch, TensorFlow, MXNet, and Paddle. The coarse-grained and fine-grained datasets have 408 and 3,270 samples, respectively. In this work, we only experiment on the coarse-grained subset.

## 4 Experiments

We present experiments of multilingual training for code translation (Section 4.1). We then introduce a novel evaluation metric **Debugging Success Rate@K** for **program-level** code translation (Section 4.2) and the first comprehensive exploration of ChatGPT for code translation (Section 4.3).

### 4.1 Multilingual Modeling

Multilingual modeling has been pivotal in broadening the applicability of neural machine translation (Aharoni et al., 2019; Wang et al., 2020; Zhu et al., 2023; Johnson et al., 2017). This is primarily evidenced in enhancing the performance of low-resource languages and cross-language transfer learning (Mohammadshahi et al., 2022; Zoph et al., 2016; Nguyen and Chiang, 2017; Johnson et al., 2017). CodeTransOcean covers nearly fifty

---

[8]AutoTransExecuter only supports translation from any source language to Python. We discuss it in Limitations.

[9]https://www.assemblyai.com/blog/pytorch-vs-tensorflow-in-2023/

[10]https://onnx.ai/

| Average | One-to-One (baseline) | Many-to-One | Many-to-Many | One-to-Many |
|---|---|---|---|---|
| High-resource | 4.68 | 5.56 (↑ 0.88) | 5.94 (↑ 1.26) | 6.18 (↑ 1.50) |
| Low-resource | 4.83 | 4.85 (↑ 0.02) | 4.95 (↑ 0.12) | 5.84 (↑ 1.01) |
| All | 5.19 | 5.31 (↑ 0.12) | 5.81 (↑ 0.62) | 6.42 (↑ 1.23) |

Table 3: Average BLEU scores of the four multilingual modeling strategies, **One-to-One**, **Many-to-One**, **Many-to-Many**, and **One-to-Many**, for All language pairs, High-resource language pairs, and Low-resource language pairs.

programming languages and deep learning frameworks. We use its datasets to explore multilingual modeling on code translation tasks.

**Experimental Setups** In this work, we use pre-trained CodeT5+ (Wang et al., 2023b)[11] as the backbone based on its superior performance on code understanding and generation evaluations reported in (Wang et al., 2023b). We use the MultilingualTrans dataset to investigate four multilingual modeling strategies based on data sharing in the source or target language or both, namely, *One-to-One*, *One-to-Many*, *Many-to-One*, and *Many-to-Many*, with One-to-One as the baseline. Details of the four strategies are in Appendix A.5. To understand the strengths and weaknesses of the four strategies, we compare their average performance on *all language pairs* and focus on *low-resource* and *high-resource pairs*. Since the CodeBLEU metric (Ren et al., 2020) does not cover all eight languages in MultilingualTrans, we use BLEU to measure translation accuracy for the four strategies. Then, we establish baselines for the DLTrans and NicheTrans datasets.

We rank the resource richness of the eight programming languages in MultilingualTrans in descending order based on their amounts in the CodeT5+ pre-training data, as Java, PHP, C, C#, Python, C++, and Go (Visual Basic is not covered by the CodeT5+ pre-training data). Based on this ranking, we consider Visual Basic, C++, and Go as low-resource languages and Java, PHP and C as high-resource languages.

**Results and Analysis** Detailed experimental results are shown in Table 14 in Appendix. For **All** language pairs, the performance of the four strategies is ranked as **One-to-Many > Many-to-Many > Many-to-One > One-to-One**. (1) Under One-to-Many strategy, the model encoder can provide more comprehensive information for source language translation due to its ability to absorb more source language features, thereby improving generalizability of the model. (2) Many-to-Many can be considered as expanding the One-to-Many strategy by employing a greater volume of non-source language data for training. Since the encoder must be attuned to the features of various languages simultaneously under Many-to-Many, parameter sharing may potentially undermine the performance. (3) Many-to-One helps the model to learn from a broader range of data than the baseline. Specific patterns or expressions in diverse source languages assist the model in more precisely comprehending how to translate into the target language. The shared semantic representations across different source languages allow the model to implement effective transfer learning strategies. Furthermore, increase in training samples enables the model to optimize the loss function more stably. These results are consistent with previous findings on multilingual modeling for natural language translation (Aharoni et al., 2019): Many-to-Many models, trained across multiple target languages instead of just one target language, can function effectively as a regularization strategy for Many-to-One, thereby reducing the possibility of over-matching.

For *High-resource* and *Low-resource* languages, as shown in Table 3, the ranking of the four strategies is the same as for *All*, but there is notable difference in their adaptability across languages of varying resource scales. High-resource languages can take advantage more effectively from the shared information across multiple source languages; whereas, low-resource languages are relatively less equipped to handle the additional uncertainty and noise introduced by shared parameters, and thus often have to rely on a larger volume of source language data to optimize their benefits.

Results from the Many-to-Many strategy on DLTrans and NicheTrans datasets are shown in Tables 4 and 5. The experimental results suggest that significant improvements in translation accuracy can be achieved by swapping the source and target languages in the training set to facilitate data augmentation and training a bidirectional model.

---

[11]We will conduct evaluations of a broader selection of models on our datasets in future work, including LLaMA (Touvron et al., 2023), WizardCoder (Luo et al., 2023), etc.

| Method | Metric | PyTorch | | | TensorFlow | | | MXNet | | | Paddle | | |
|---|---|---|---|---|---|---|---|---|---|---|---|---|---|
| | | EM | BLEU | CodeBLEU | EM | BLEU | CodeBLEU | EM | BLEU | CodeBLEU | EM | BLEU | CodeBLEU |
| Naive | PyTorch | – | – | – | 27.27 | 66.25 | 69.46 | 28.18 | 72.77 | 76.63 | 30.91 | 80.35 | 83.13 |
| | TensorFlow | 27.27 | 66.32 | 68.92 | – | – | – | 29.09 | 63.79 | 67.94 | 27.27 | 63.04 | 65.81 |
| | MXNet | 28.18 | 72.86 | 74.15 | 29.09 | 63.84 | 66.06 | – | – | – | 28.18 | 69.49 | 71.09 |
| | Paddle | 30.91 | 80.25 | 84.83 | 27.27 | 62.94 | 67.78 | 28.18 | 69.43 | 75.09 | – | – | – |
| CodeT5+ | PyTorch | – | – | – | 35.45±0.91 | 71.16±0.73 | 70.54±0.75 | 42.73±2.41 | 81.76±0.45 | 82.52±0.56 | 43.64±1.58 | 85.76±0.60 | 85.07±0.74 |
| | TensorFlow | 34.85±1.38 | 71.97±0.56 | 71.08±0.72 | – | – | – | 36.67±1.89 | 72.77±0.61 | 73.04±0.18 | 29.70±2.63 | 69.38±0.38 | 68.76±0.32 |
| | MXNet | 32.12±2.29 | 77.79±0.13 | 76.43±0.14 | 31.82±1.58 | 67.22±0.39 | 67.68±0.27 | – | – | – | 29.09±0.91 | 74.26±0.46 | 73.27±0.42 |
| | Paddle | 43.03±4.10 | 86.25±0.86 | 86.09±0.88 | 29.39±2.93 | 69.43±0.57 | 69.57±0.51 | 35.75±0.53 | 78.65±0.62 | 79.46±0.38 | – | – | – |

Table 4: Results on DLTrans of Naive and CodeT5+_220M with **Many-to-Many** strategy. We run each experiment with 3 random seeds and report the mean and standard deviation of EM, BLEU, and CodeBLEU scores.

| BLEU | Naive | Two-way | One-way |
|---|---|---|---|
| Many-to-C | 2.36 | 4.60 | 4.86 |
| Many-to-C# | 2.53 | 4.48 | 3.82 |
| Many-to-C++ | 1.99 | 4.78 | 3.32 |
| Many-to-Go | 3.11 | 5.24 | 3.19 |
| Many-to-Java | 3.18 | 5.23 | 5.34 |
| Many-to-PHP | 4.37 | 2.46 | 1.98 |
| Many-to-Python | 2.87 | 2.38 | 1.67 |
| Many-to-VB | 1.69 | 2.17 | 1.97 |
| Average | 2.76 | **3.92** | 3.27 |

Table 5: BLEU scores on NicheTrans of Naive and CodeT5+_220M with **Many-to-Many** strategy. **One-way** denotes training models only from niche to popular, while **Two-way** denotes training in both directions.

Notably, prior studies on multilingual neural machine translation often overlook the comparison between One-to-Many and other strategies. Nevertheless, One-to-Many demonstrates superiority over the One-to-One baseline across all our experiments. Overall, our results strongly recommend a targeted multilingual modeling strategy for code translation, as it not only can translate multiple language pairs with a single model, but also achieves better and more stable accuracy than baselines.

## 4.2 Debugging Success Rate@K

For evaluations, we adopt existing code translation evaluation metrics in our experiments, including **Exact Match (EM)**, **BLEU**, and **CodeBLEU** (details are in Appendix A.1.2). However, all these metrics are based on surface-form matching (or with some adaptations as for CodeBLEU) and are not suitable for our **program-level** translation tasks since they cannot reliably evaluate functional correctness of translated code. Moreover, in real-world software development scenarios, developers typically ensure the functionality of code by testing and debugging upon completion, rather than writing and testing multiple versions of the code to achieve the expected functionality as measured by the existing pass@k (Kulal et al., 2019) metric.

Meanwhile, recent research shows that LLMs such as ChatGPT demonstrate preliminary code debugging capabilities (Chen et al., 2023b,a). Hence, we propose a novel and robust evaluation metric for LLM on code translation, **Debugging Success Rate@K (DSR@K)**, by measuring whether the translated code can be compiled and executed with the same behavior as the input source code, with K rounds of debugging. **To the best of our knowledge, *DSR@K* is the first metric designed to accurately reflect real-world software development scenarios.**

*DSR@K* is the percentage of the samples that successfully execute and produce the expected results among all samples. Each sample is given $K$ generation and debugging attempts by an LLM. If the generated code successfully executes and produces the expected results with these $K$ rounds, the sample is marked as successful. *DSR@K* is computed as $\frac{1}{N}\sum_{i=1}^{N} S(i, K)$, where $N$ denotes the total number of samples. If the $i^{th}$ code sample succeeds within $K$ attempts, then $S(i, K) = 1$; otherwise, $S(i, K) = 0$. Note that DSR@0 can be used for program-level code translation evaluation for any models. In this work, we employ DSR@K to evaluate the ability of LLMs such as ChatGPT for debugging code and translating code with debugging results.

## 4.3 ChatGPT for Code Translation

The recent LLM ChatGPT demonstrates competitive performance on language generation tasks such as summarization and machine translation (Yang et al., 2023; Peng et al., 2023; Gao et al., 2023). However, ChatGPT for code translation has not been systematically explored. We study the effectiveness and potential of ChatGPT on code translation and investigate strategies to improve its performance. We use **DSR@K** as the principal evaluation metric since we focus on the practical usability of ChatGPT. We use the ChatGPT API and gpt-3.5-turbo as the default model and evaluate on the

**LLMTrans** dataset for all experiments. We investigate the efficacy of prompts and hyperparameters and context in zero-shot setting, then compare one-shot versus zero-shot and study Chain-of-Thought.

**Effect of Prompts and Hyperparameters** Prior works show that prompts can influence the performance of ChatGPT (Zhong et al., 2023; Peng et al., 2023; Jiao et al., 2023). We set an initial prompt "Translate [SL] to [TL]:[SC]." as the baseline, where [SL] and [TL] denote the source language and the target language respectively and [SC] denotes the source code. We also add "Do not return anything other than the translated code." for each prompting strategy to require ChatGPT to return only code in order to ease code execution. We design three prompt variants. Details of the experimental settings and prompt variants are in Appendix A.6. We also investigate the effect of hyperparameters on code translation performance.

As shown in Table 6, implementing role assignments, clarifying usage, and polite inquiry in prompts all degrade the performance compared to the baseline prompt. These results show that the baseline with the most straightforward prompt produces the best performance, possibly because it provides clear, short, and unambiguous instructions for the task to the model. More intricate prompting strategies may introduce noise and confuse ChatGPT. The performance of polite inquiry prompt is comparable to but still worse than the baseline performance. We speculate that the improvement from polite inquiries in prior studies (Akın, 2023) may stem from their explicit and comprehensive formulations which make it easier for the model to understand the task requirements. We also observe in Table 6 that same as prior findings, BLEU and CodeBLEU have no obvious positive correlations with the debugging success rate (DSR@0). Since the reference target code exhibits the same functionality as the source language code but their execution results could differ slightly, EM also does not correlate with DSR@0. Therefore, in subsequent experiments, we only report DSR@0. We also evaluate the CodeT5+_220M model on LLMTrans with the Many-to-Many strategy and find that DSR@0 is 0, suggesting that CodeT5+_220M Zero-shot is unable to generate executable translation results.

ChatGPT selects the token with the highest probability during generation. The hyperparameter *temperature* influences the randomness of the generated text, while *top_p* controls the range of vocabu-

| Strategy | Expt #num | EM | BLEU | CodeBLEU | DSR@0 |
|---|---|---|---|---|---|
| Baseline | – | 0.29 | 10.83 | 24.46 | **48.57%** |
| Role assignments | 1 | 0.00 | **11.06** | 24.36 | 43.43% |
| | 2 | 0.00 | **11.06** | 24.48 | 43.14% |
| | 3 | 0.00 | 10.70 | 24.08 | 41.71% |
| | 4 | 0.00 | 10.73 | 24.08 | 40.86% |
| Polite inquiry | 1 | 0.29 | 10.83 | 24.37 | **47.71%** |
| | 2 | **0.86** | 10.87 | 24.26 | **47.71%** |
| Clarify usage | – | 0.29 | 10.63 | 24.11 | 44.00% |
| Divide-and-Conquer | – | 0.00 | 7.44 | **25.30** | 22.86% |

Table 6: Zero-shot performance of ChatGPT with different prompt variants and contextual strategies. Baseline denotes ChatGPT with the baseline prompt. Details of the prompt variants (Expt #num) are in Appendix A.6.

| $K^{th}$ Debug | DSR | $K^{th}$ Debug | DSR |
|---|---|---|---|
| 0 | 48.57% | 2 | 52.29% |
| 1 | 51.43% | 3 | **52.57%** |

Table 7: ChatGPT performance at the $K^{th}$ debugging.

lary considered during generation. Higher temperature or top_p could increase diversity in the generated results from ChatGPT. However, as shown in Table 16 in Appendix, independently varying temperature or top_p does not notably change the performance of ChatGPT; hence for the other ChatGPT experiments, we set both temperature and top_p as 0 to ensure stability an reproducibility.

**Effect of Context** We explore a *Divide-and-Conquer* strategy, which segments the source language code into snippets (e.g., functions and subfunctions), translate each snippet independently, then merge their outputs as the final result. As shown in Table 6, Divide-and-Conquer significantly degrades the performance. We hypothesize that lack of the global context in Divide-and-Conquer could prevent ChatGPT from considering the overall structure and variable configurations of the code for translation.

**Effect of Self-debugging** Since ChatGPT has shown preliminary capability in error detection and correction during code generation (Shinn et al., 2023; Chen et al., 2023b; Kim et al., 2023; Nair et al., 2023; Madaan et al., 2023), we use ChatGPT to perform multiple rounds of self-debugging and investigate the impact on DSR. Specifically, ChatGPT first translates the source language code into the target language (which is Python as in our AutoTransExecuter) and then attempts to execute the translated code. If the execution passes and executing the translated code exhibits the same functionality as the source code, it is regarded as

a successful execution. Otherwise, feedback from the compiler will be also fed to ChatGPT for the next round of translation, and this process is repeated until reaching a pre-defined number $K$ of debugging rounds. The whole process is shown in Table 17 in Appendix. As shown in Table 7, DSR improves significantly with multiple rounds of self-debugging. The first self-debugging improves DSR by **3%** absolutely. Each subsequent round of self-debugging brings further gain but DSR begins to plateau after the second debugging round. This suggests that ChatGPT has limitations in its capacity to rectify errors after multiple debugging cycles, which is consistent with human behaviors.

**Effect of One-shot**   In-context learning (Brown et al., 2020) allows the model to learn from input examples, enabling it to understand and manage each new task. This method has been validated as an effective strategy for enhancing the performance of model inference (Peng et al., 2023; Liu et al., 2023a). Therefore, we explore one-shot learning for ChatGPT on code translation. We investigate three one-shot learning sample selection strategies. Descriptions of the strategies and the corresponding prompts are in Appendix A.7.

Table 8 shows that all three One-shot learning strategies effectively improve DSR@0 of ChatGPT over the Zero-shot baseline. The Experiment#2 strategy (provided contextual example has both same source and target languages as the original task) achieves the best performance, yielding **1.72%** absolute gain in DSR@0, with Experiment #1 (example has the same target language but different source language) and #3 (example has different source and target languages) following closely with 1.14% and 0.29% absolute gains, respectively. These results show that One-shot learning entirely tailored to the translation requirements is most effective in boosting code translation performance for ChatGPT. The results corroborate previous findings in natural language translation (Peng et al., 2023) that the performance of ChatGPT is sensitive to the provided contextual example in One-shot learning.

**Effect of Chain-of-Thought**   Chain-of-Thought (CoT) allows the model to simulate an orderly and structured way of thinking by sorting out the thinking process. It helps guide the model to output the final answer step by step (Wei et al., 2022; Peng et al., 2023; Kojima et al., 2022). For code translation, we investigate four CoT strategies. Detailed

| Strategy | Expts #num | DSR@0 | Strategy | Expts #num | DSR@0 |
|----------|------------|-------|----------|------------|-------|
| Baseline | – | **48.57%** |     | 1 | 46.00% |
|          | 1 | 49.71% | CoT | 2 | 42.57% |
| One-shot | 2 | **50.29%** |     | 3 | **48.29%** |
|          | 3 | 48.86% |     | 4 | 45.43% |

Table 8: Performance of ChatGPT with One-shot and CoT strategies compared to the Zero-shot Baseline. Details of Expt #num are in Appendix A.7 and A.8.

descriptions and translation prompts for each strategy are in Appendix A.8. As shown in Table 8, CoT degrades executability of the translated code. In Experiment #2, DSR@0 even declines by 6% absolutely. We study the translation results of ChatGPT and find that when CoT strategies are applied, the model tends to translate the source code line by line, neglecting compatibility issues between libraries and functions in different languages. CoT also compromises the global planning ability of the model. These observations are consistent with the findings in (Peng et al., 2023) that CoT may lead to word-by-word translations of natural language, thereby degrading the translation quality.

**Fuzzy Execution**   To address the limitations of existing evaluation metrics and our AutoTransExecuter, we propose another novel code translation evaluation metric **fuzzy execution** using LLMs in Section Limitations, inspired by recent progress in using LLMs as evaluation metrics for NLP tasks. Our preliminary studies evaluates the performance of ChatGPT for predicting whether a given code can be executed or not, and if executable, also for predicting the executed output. Experimental results show that using ChatGPT for fuzzy execution is not yet practical and demands future research.

## 5   Conclusion

We construct CodeTransOcean, a comprehensive code translation benchmark that includes multilingual and cross-framework datasets. We demonstrate that multilingual modeling has remarkable potential in enhancing code translation quality. We also reveal the superior code translation capability of ChatGPT and advanced strategies lead to significant performance gains. Moreover, we introduce fuzzy execution that may overcome limitations of existing metrics but requires future research. In summary, we provide a comprehensive suite of resources, tools, and baselines for code translation.

## 6 Limitations

Existing match-based evaluation metrics for code translation (Papineni et al., 2002; Ren et al., 2020; Eghbali and Pradel, 2022; Zhou et al., 2023; Tran et al., 2019) focus solely on semantics, overlooking executability of the code and the functional equivalence under different implementations. Execution-based metrics (Kulal et al., 2019; Hao et al., 2022; Hendrycks et al., 2021; Rozière et al., 2020; Dong et al., 2023) that require providing test cases are expensive to conduct in practice, and the significant overhead of executing numerous test cases and the heightened security risks during the execution process remain unresolved. It is crucial to establish an evaluation metric that overcomes these limitations.

Our proposed DSR@K and the automated Auto-TransExecuter aim to measure the executability of the code and reflect the real-world software development scenarios. However, AutoTransExecuter currently only supports Python as the target language. This is mainly due to the fact that different programming languages necessitate distinct runtime environments and libraries, making it particularly challenging to automatically detect and install the required dependencies for each code. While certain existing tools, such as Dynatrace[12], can carry out dependency detection, the range of supported programming languages remains limited. Moreover, the configuration methods for compilers vary substantially among different programming languages, which further complicates automated configuration. In addition, fully automated execution systems could be exploited by malicious code, thus necessitating further security measures. Therefore, achieving this goal requires overcoming many technical and practical difficulties.

To address limitations of existing evaluation metrics and limitations of AutoTransExecuter, we propose another novel code translation evaluation metric **fuzzy execution**.

Recent studies have begun to utilize LLMs as evaluation metrics in the field of NLP (Chen et al., 2023c; Wang et al., 2023a; Fu et al., 2023; Kocmi and Federmann, 2023; Ji et al., 2023). Inspired by these works, we create a new dataset **ExecuteStatus** by randomly selecting 300 executable samples from MultilingualTrans and 300 non-executable samples from the translation results of ChatGPT.

---

[12] https://www.dynatrace.com/platform/artificial-intelligence/dependency-detection/

| Zero-Shot | | | | Few-Shot | | | |
|---|---|---|---|---|---|---|---|
| TN | FP | FN | TP | TN | FP | FN | TP |
| 292 | 8 | 238 | 62 | 294 | 4 | 242 | 58 |
| ✓12 | ×280 | | | ✓14 | ×282 | | |

Table 9: Confusion matrix of fuzzy execution prediction by ChatGPT with Zero-shot and Few-shot settings.

| Metrics | Calculation formula | Zero-Shot | Few-Shot |
|---|---|---|---|
| Accuracy | $\dfrac{TP + TN}{TP + TN + FN + FP}$ | 59.00% | 59.00% |
| Precision | $\dfrac{TP}{TP + FP}$ | 88.57% | 93.55% |
| Recall | $\dfrac{TP}{TP + FN}$ | 20.67% | 19.33% |
| F1 scores | $2 \cdot \dfrac{Precision \cdot Recall}{Precision + Recall}$ | 33.52% | 32.04% |

Table 10: Performance of ChatGPT on predicting fuzzy execution.

Each entry in this dataset includes the execution status and, if executable, the result of the execution. We use ExecuteStatus and AutoTransExecuter to evaluate the performance of ChatGPT for predicting whether a given code can be executed or not, and if executable, also predict the executed output. The Zero-shot prompts are shown in Table 18 in Appendix. For the Few-shot strategy, in addition to the Zero-shot baseline, we include an example of executable code and an example of non-executable code, as detailed in Table 18.

We define fuzzy execution as first testing the consistency between the actual pass rate and the predicted pass rate of ChatGPT, followed by further testing the accuracy in predicting execution results using ChatGPT without relying on a compiler. Since we are interested in the ability of Chat-GPT to identify samples that cannot actually be executed accurately, we present the confusion matrix in Table 9 based on the results. To evaluate the performance of ChatGPT on the fuzzy execution prediction task, we use the standard accuracy, precision, recall, and F1 scores. Experimental results based on these evaluation metrics are in Table 10. The low accuracy, recall and F1 scores show that ChatGPT still has difficulty in identifying errors in the code, exhibiting about an 88% tendency to predict that the code is executable. Overall, ChatGPT has low accuracy in the binary classification task of "whether it can be executed", and its ability to predict execution results, being at a scant 4%, clearly

requires further enhancement. Thus, using Chat-GPT for fuzzy execution is not yet practical (Liu et al., 2023b). Despite this, fuzzy execution with LLMs holds the potential to overcome the deficiencies of current code translation evaluation metrics. We will continue this exploration in future work.

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

# A Appendix

## A.1 Related Work

### A.1.1 Code Translation Methods

**Naive Copy** directly duplicates the source code as the target code without making any modifications. Given that the results produced by this method are often unusable, it is treated as the lower bound of performance for code translation. Early code translation relies heavily on manual rewriting, which requires developers to have a deep understanding of both source and target languages along with the ability to navigate various complex programming structures and semantic challenges. This method is inefficient, costly, and prone to errors.

Automatic code translation methods fall into several categories. **Compilers and transpilers**[13] can automatically translate the source code into a target language, significantly saving time and effort. However, these methods cannot fully preserve all the linguistic features and behaviors of the source code, nor can they comprehend the intent and semantics inherent to the source code as humans do. **Rule-based methods** (Weisz et al., 2021, 2022; Rozière et al., 2020) treat the code translation task as a program synthesis problem. They define a set of transformation rules and employ the rules or pattern matching for code translation. Research on rule-based methods is quite scarce, mainly because they overly rely on the completeness of the rules and also require a considerable amount of manual preprocessing.

**Neural network based methods** have become dominant in the field of code translation in recent

---

[13] https://en.wikipedia.org/wiki/Source-to-source_compiler

years. These methods mainly treat code translation as a sequence-to-sequence generation problem. Among them, Chen et al. (Chen et al., 2018) are the first to successfully apply neural networks to code translation, designing a tree-to-tree neural model. CodeBERT (Feng et al., 2020) significantly improves code translation accuracy by pretraining models with masked language modeling and replaced token detection. GraphCode-BERT (Guo et al., 2021) further improves code translation accuracy by introducing two additional pre-training tasks as edge prediction and node alignment. CodeT5 (Wang et al., 2021), based on the Transformer encoder-decoder architecture, achieves excellent performance on code translation through four pre-training tasks, namely, masked span prediction, identifier tagging, masked identifier prediction, and bimodal dual generation. With a similar architecture as CodeT5, PLBART (Ahmad et al., 2021) adopts three tasks of token masking, token deletion and token infilling for denoising seq2seq pre-training, which enables PLBART to infer language syntax and semantics and to learn how to generate language coherently. Nat-Gen (Chakraborty et al., 2022) forces the model to learn to capture intent of the source code by setting up "Code-Naturalization" tasks during pre-training, and forces the model to make the generated code closer to the human-written style.

In the line of neural network based methods, recently released **large language models (LLMs)** (e.g., ChatGPT (OpenAI, 2023)) have shown remarkable performance in a wide range of NLP tasks with instructions and a few in-context examples. ChatGPT is built upon GPT and is optimized with Reinforcement Learning from Human Feedback. ChatGPT can efficiently understand and generate code sequences, and can self-learn from human feedback to improve the quality and accuracy of its outputs. This significant advancement has markedly propelled progress in the field of code translation.

### A.1.2 Code Translation Metrics

**Match-Based Evaluation Metrics** These evaluation metrics are based on the similarity between the translation output and the reference translation. Among them, the Exact Match (EM) metric calculates the percentage of translation outputs that *exactly* match the reference translation, which overlooks the fact that the same function can be implemented in various ways. The Bilingual Evaluation Understudy (BLEU) (Papineni et al., 2002) metric evaluates the similarity between the translation output and the reference translation by multiplying the geometric average of n-gram precision scores with a brevity penalty. The CodeBLEU (Ren et al., 2020) metric extends BLEU by considering syntactic and semantic characteristics of programming languages; it not only considers shallow matching but also pays attention to syntactic and semantic matching. CrystalBLEU (Eghbali and Pradel, 2022) focuses more on the inherent differences between source code and natural language, such as trivial shared n-gram syntax. Code-BERTScore (Zhou et al., 2023) uses pre-trained models to encode the translation output and reference translation, then calculates the dot product similarity between them, enabling comparisons of code pairs with distinct lexical forms. However, CodeBLEU, CrystalBLEU, and CodeBERTScore have limitations as they only support a limited range of programming languages and cannot be used in general multilingual scenarios. Ruby (Tran et al., 2019), a new method for evaluating code translation, considers the lexical, syntactic, and semantic representations of source code. However, its codebase has not yet been open-sourced. These match-based evaluation metrics can only evaluate the surface form and semantic differences of the code, while neglecting the executability of the code and the functional equivalence of implementation variations.

**Execution-Based Evaluation Metrics** Execution-based evaluation metrics mainly compare the executed result of the generated code with the expected result. The PASS@k score (Kulal et al., 2019) is evaluated by unit tests: if any of the $k$ samples meets the expected result, the generated result is deemed successful. AvgPassRatio (Hao et al., 2022; Hendrycks et al., 2021) evaluates the overall executable result of code by calculating the average pass rate of test cases. Computational accuracy (Rozière et al., 2020) measures the quality of the generated code snippet by comparing the output of this snippet with the reference code snippet when given the same input. Additionally, CodeScore (Dong et al., 2023) claims that it can estimate the PassRatio of test cases for the generated code without executing the code, but its codebase has not yet been open-sourced. These execution-based evaluation metrics require construction of executable test

| Method | Train | Dev | Test | Total | Method | Train | Dev | Test | Total |
|---|---|---|---|---|---|---|---|---|---|
| C ↔ C# | 796 | 84 | 169 | 1049 | C ↔ C++ | 799 | 149 | 298 | 1246 |
| C ↔ Go | 877 | 227 | 454 | 1558 | C ↔ Java | 813 | 171 | 343 | 1327 |
| C ↔ Python | 901 | 213 | 426 | 1540 | C ↔ PHP | 296 | 51 | 102 | 449 |
| C ↔ VB | 617 | 97 | 194 | 908 | C++ ↔ C# | 748 | 79 | 160 | 987 |
| C++ ↔ Go | 792 | 208 | 418 | 1418 | C++ ↔ Java | 753 | 172 | 345 | 1270 |
| C++ ↔ Python | 842 | 202 | 405 | 1449 | C++ ↔ PHP | 291 | 53 | 106 | 450 |
| C++ ↔ VB | 586 | 97 | 195 | 888 | C# ↔ Go | 777 | 100 | 202 | 1079 |
| C# ↔ Java | 750 | 86 | 174 | 1010 | C# ↔ Python | 813 | 99 | 199 | 1111 |
| C# ↔ PHP | 293 | 40 | 80 | 413 | C# ↔ VB | 597 | 70 | 142 | 809 |
| Java ↔ Go | 793 | 221 | 443 | 1457 | Java ↔ Python | 838 | 217 | 436 | 1491 |
| Java ↔ PHP | 574 | 119 | 239 | 932 | Java ↔ VB | 610 | 104 | 210 | 924 |
| Go ↔ Python | 887 | 314 | 628 | 1828 | Go ↔ PHP | 606 | 128 | 258 | 992 |
| Go ↔ VB | 618 | 116 | 232 | 966 | PHP ↔ Python | 927 | 185 | 370 | 1482 |
| PHP ↔ VB | 267 | 44 | 88 | 399 | VB ↔ Python | 644 | 114 | 229 | 987 |

Table 11: Composition and Distribution of the Multilingual Dataset. The numbers refer to the number of code pair samples. VB is short for Visual Basic. [Return to Section 3.1]

sets, which could be costly. Furthermore, due to potential security threats from the execution environment and the code, they need to be run in an isolated sandbox.

## A.2 Data Management

### A.2.1 Data Sources & Licenses

We collect CodeTransOcean from two different platforms. The MultilingualTrans and NicheTrans datasets are collected from Rosetta Code[14], a programming site presenting solution strategies for identical tasks across as many programming languages as possible, thereby demonstrating both similarities and differences among these languages. We strictly adhere to the data distribution license of the platform as Attribution-ShareAlike 4.0 International (CC BY-SA 4.0) license[15].

The DLTrans dataset is derived from an open-source teaching platform *Dive into Deep Learning*[16], which is dedicated to teaching deep learning knowledge ranging from theoretical background, conceptual understanding, to coding practices. We strictly adhere to the data distribution license of this platform as Apache-2.0[17]. To ensure legal and regulated use of these datasets, we require strict adherence to these licenses.

### A.2.2 Data Processing

**Multilingual Datasets** Given the variations in compilation requirements among programming languages, we keep the original format as much as possible to ensure the compilability of the data while ensuring its accuracy. Additionally, we employ a duplicate-file-detection tool to identify and remove duplicate data from the dataset to avoid any potential data leakage problems during model training.

**Cross-Framework Dataset** To ensure the compilability of Python, we keep the original formatting information. We manually verify all automatically collected samples, identify and exclude samples that do not meet the requirements.

### A.2.3 Data Quality

We randomly select 1K samples from each dataset within CodeTransOcean for manual quality assessment. We find that for the MultilingualTrans dataset with compilation requirements, the compilability rate exceeds **90%**. We verify that the code pairs in each dataset are functionally identical and confirm that CodeTransOcean is of high quality.

Additionally, during data collection, we pay special attention to the diversity of domain knowledge and code styles. CodeTransOcean includes various code examples ranging from basic syntactic structures to complex algorithm implementations, as well as building neural networks from scratch and conducting training and inference. This rich

[14]https://rosettacode.org/wiki/Rosetta_Code
[15]https://creativecommons.org/licenses/by-sa/4.0/
[16]https://github.com/d2l-ai/d2l-zh
[17]https://github.com/d2l-ai/d2l-zh/blob/master/LICENSE

diversity ensures that CodeTransOcean reflects a wide variety of real-world scenarios.

## A.3 Specific Challenges in Implementing Cross-framework Translation

Firstly, there are significant design differences between frameworks, including data processing methods, model-building strategies, and network connection techniques. Secondly, the inherent complexity of DL code increases the difficulty of conversion, as these codes usually contain various components such as neural network layers, loss functions, optimizers, and learning rate schedulers. Thirdly, there are significant inconsistencies in the code structure of different frameworks, such as code organization and variable naming rules. Lastly, cross-platform compatibility must be considered because DL code may encounter compatibility issues when executing on different hardware platforms (e.g., GPUs, CPUs, TPUs) and operating systems.

## A.4 Code Examples on Different Deep Learning Frameworks

Figures 1 and 2 show the implementation of two different deep learning components in various deep learning frameworks.

## A.5 Multilingual Modeling

**One-to-One** For each language pair in the dataset, we train an independent model, e.g., translating C++ to Java.

**One-to-Many** We train individual models from one language to many other languages, e.g., translating Python to all other languages.

**Many-to-One** We train individual models from multiple languages to one language, e.g., translating all other languages to Python.

**Many-to-Many** We train a unified model for the multiple to multiple languages in the dataset, which can handle translations between all languages.

We ensure all experiments are performed under the same hyperparameters and environment for comparison. Table 13 shows these in detail.

## A.6 Prompt Variations

**Role Assignment** (Peng et al., 2023; AlKhamissi et al., 2023; Wu et al., 2023; Akın, 2023) We configured two distinct roles for the model, each with unique skills. This arrangement empowers the

**PyTorch**

```
1  class MaskedSoftmaxCELoss(nn.CrossEntropyLoss):
2      def forward(self, pred, label, valid_len):
3          weights = torch.ones_like(label)
4          weights = sequence_mask(weights, valid_len)
5          self.reduction='none'
6          unweighted_loss = super(MaskedSoftmaxCELoss, self
                ).forward(
7              pred.permute(0, 2, 1), label)
8          weighted_loss = (unweighted_loss * weights).mean(
                dim=1)
9          return weighted_loss
```

**TensorFlow**

```
1   class MaskedSoftmaxCELoss(tf.keras.losses.Loss):
2       def __init__(self, valid_len):
3           super().__init__(reduction='none')
4           self.valid_len = valid_len
5
6       def call(self, label, pred):
7           weights = tf.ones_like(label, dtype=tf.float32)
8           weights = sequence_mask(weights, self.valid_len)
9           label_one_hot = tf.one_hot(label, depth=pred.
                shape[-1])
10          unweighted_loss = tf.keras.losses.
                CategoricalCrossentropy(
11              from_logits=True, reduction='none')(
                    label_one_hot, pred)
12          weighted_loss = tf.reduce_mean((unweighted_loss*
                weights), axis=1)
13          return weighted_loss
```

**MXNet**

```
1   class MaskedSoftmaxCELoss(gluon.loss.SoftmaxCELoss):
2       def forward(self, pred, label, valid_len):
3           weights = np.expand_dims(np.ones_like(label),
                axis=-1)
4           weights = npx.sequence_mask(weights, valid_len,
                True, axis=1)
5           return super(MaskedSoftmaxCELoss, self).forward(
                pred, label, weights)
```

**Paddle**

```
1  class MaskedSoftmaxCELoss(nn.CrossEntropyLoss):
2      def forward(self, pred, label, valid_len):
3          weights = paddle.ones_like(label)
4          weights = sequence_mask(weights, valid_len)
5          self.reduction='none'
6          unweighted_loss = super(MaskedSoftmaxCELoss, self
                ).forward(
7              pred, label)
8          weighted_loss = (unweighted_loss * weights).mean(
                axis=1)
9          return weighted_loss
```

Figure 1: *Softmax cross-entropy loss function with masking* in PyTorch, TensorFlow, MXNet and Paddle.

model to simulate more domain-adaptable and specialized expert roles.

**Polite inquiry** (Akın, 2023) These strategies add polite expression and set up imperative and interrogative requests. Given that ChatGPT is designed to simulate human conversation styles as closely as possible, including understanding and simulating polite language expressions. Therefore, we expect these strategies to boost the comprehension of the model and augment the quality of its generated results.

**Clarify usage** This strategy aims to make the model clearly aware of its requirements during the code translation process - the generated code needs to be guaranteed to execute without issues.
The translation prompts of the above four strategies

| Method | Train | Dev | Test | Total | Method | Train | Dev | Test | Total |
|---|---|---|---|---|---|---|---|---|---|
| Ada | 6022 | 464 | 937 | 7423 | Elixir | 3618 | 297 | 599 | 4514 |
| Arturo | 3802 | 470 | 947 | 5219 | Erlang | 3269 | 217 | 449 | 3935 |
| AutoHotKey | 4305 | 555 | 1120 | 5980 | Factor | 4724 | 860 | 1756 | 7340 |
| AWK | 3880 | 578 | 1162 | 5620 | Forth | 3619 | 339 | 690 | 4648 |
| BBC Basic | 3663 | 239 | 485 | 4387 | Fortran | 4050 | 305 | 617 | 4972 |
| Clojure | 3617 | 310 | 633 | 4560 | Groovy | 3773 | 227 | 467 | 4467 |
| Common Lisp | 5085 | 467 | 933 | 6485 | Haskell | 5647 | 1045 | 2097 | 8789 |
| D | 5541 | 825 | 1662 | 8028 | Icon | 2646 | 158 | 326 | 3130 |
| Delphi | 3547 | 436 | 889 | 4872 | J | 5476 | 1204 | 2422 | 9102 |
| Julia | 5829 | 1511 | 3055 | 10395 | Ruby | 5793 | 1290 | 2600 | 9683 |
| Lua | 5316 | 677 | 1366 | 7359 | COBOL | 2438 | 167 | 355 | 2960 |
| Mathematica | 5485 | 1046 | 2105 | 8636 | REXX | 5595 | 1118 | 2241 | 8954 |
| MATLAB | 2872 | 157 | 322 | 3351 | R | 2803 | 197 | 402 | 3402 |
| Nim | 5814 | 1321 | 2675 | 9810 | Racket | 5646 | 901 | 1817 | 8364 |
| OCaml | 4286 | 405 | 817 | 5508 | Rust | 5146 | 717 | 1439 | 7302 |
| Pascal | 3393 | 465 | 942 | 4800 | Tcl | 5354 | 740 | 1502 | 7596 |
| Perl | 5818 | 1445 | 2914 | 10177 | PowerShell | 3563 | 240 | 490 | 4293 |
| Scala | 5852 | 1074 | 2164 | 9090 | F# | 4517 | 638 | 1287 | 6442 |
| Swift | 3653 | 404 | 818 | 4875 | | | | | |

Table 12: The number of code samples for each language in the NicheTrans datasets. [Return to Section 3.1]

| | MultilingualTrans | NicheTrans | DLTrans |
|---|---|---|---|
| Learning rate | 3e-5 | 2e-5 | 3e-5 |
| Beam size | 1 | 1 | 5 |
| Max source length | 1536 | 1536 | 512 |
| Max target length | 1536 | 1536 | 512 |
| Batch size | | 16 | |
| Max epoch | | 5 | |
| Fp16 | | True | |
| GPU | | NVIDIA Tesla V100 32GB | |

Table 13: Parameters and hardware configuration for training CodeT5+_220M (220M is the model size) on CodeTransOcean.

are shown in Table 15.

## A.7 One-Shot

**Experiment #1** selects a training sample from a high-resource language code pair as an example. In this case, the target language type aligns with the target language type of the translation request, but the source language does not.

**Experiment #2** selects a code pair whose source and target language directions are congruent with the translation requirements as an example. That is, the source and target languages of the example dynamically adjust following the translation requirements.

**Experiment #3** randomly selects a code pair as an example, in which neither the source nor the target languages match the translation requirements. The specific translation prompts are shown in Table 19.

## A.8 Chain of Thought

**Experiment #1** First, describe the function of the source code in the natural language, then translate it according to the source code and the corresponding natural language description.

**Experiment #2** First, let ChatGPT understand the function of the source code, followed by the translation, while ensuring that the function of the code remains unchanged during the translation process.

**Experiment #3** First, let ChatGPT understand the function of the source code, then predict the output result of the source code, and finally perform the translation, demanding that the translated code successfully executes.

**Experiment #4** Building upon Experiment #3 and the one-shot approach of Experiment #2, we introduce a CoT one-shot variation. That is, first, provide a case in the same direction for ChatGPT reference, then require it to understand the func-

## PyTorch

```
1  class Seq2SeqDecoder(d2l.Decoder):
2      def __init__(self, vocab_size, embed_size,
           num_hiddens, num_layers, dropout=0, **kwargs):
3          super(Seq2SeqDecoder, self).__init__(**kwargs)
4          self.embedding = nn.Embedding(vocab_size,
               embed_size)
5          self.rnn = nn.GRU(embed_size + num_hiddens,
               num_hiddens, num_layers, dropout=dropout)
6          self.dense = nn.Linear(num_hiddens, vocab_size)
7
8      def init_state(self, enc_outputs, *args):
9          return enc_outputs[1]
10
11     def forward(self, X, state):
12         X = self.embedding(X).permute(1, 0, 2)
13         context = state[-1].repeat(X.shape[0], 1, 1)
14         X_and_context = torch.cat((X, context), 2)
15         output, state = self.rnn(X_and_context, state)
16         output = self.dense(output).permute(1, 0, 2)
17         return output, state
```

## TensorFlow

```
1  class Seq2SeqDecoder(d2l.Decoder):
2      def __init__(self, vocab_size, embed_size,
           num_hiddens, num_layers, dropout=0, **kwargs):
3          super().__init__(**kwargs)
4          self.embedding = tf.keras.layers.Embedding(
               vocab_size, embed_size)
5          self.rnn = tf.keras.layers.RNN(tf.keras.layers.
               StackedRNNCells(
6              [tf.keras.layers.GRUCell(num_hiddens, dropout
                   =dropout)
7              for _ in range(num_layers)]),
                   return_sequences=True, return_state=
                   True)
8          self.dense = tf.keras.layers.Dense(vocab_size)
9
10     def init_state(self, enc_outputs, *args):
11         return enc_outputs[1]
12
13     def call(self, X, state, **kwargs):
14         X = self.embedding(X)
15         context = tf.repeat(tf.expand_dims(state[-1],
               axis=1), repeats=X.shape[1], axis=1)
16         X_and_context = tf.concat((X, context), axis=2)
17         rnn_output = self.rnn(X_and_context, state, **
               kwargs)
18         output = self.dense(rnn_output[0])
19         return output, rnn_output[1:]
```

## MXNet

```
1  class Seq2SeqDecoder(d2l.Decoder):
2      def __init__(self, vocab_size, embed_size,
           num_hiddens, num_layers, dropout=0, **kwargs):
3          super(Seq2SeqDecoder, self).__init__(**kwargs)
4          self.embedding = nn.Embedding(vocab_size,
               embed_size)
5          self.rnn = rnn.GRU(num_hiddens, num_layers,
               dropout=dropout)
6          self.dense = nn.Dense(vocab_size, flatten=False)
7
8      def init_state(self, enc_outputs, *args):
9          return enc_outputs[1]
10
11     def forward(self, X, state):
12         X = self.embedding(X).swapaxes(0, 1)
13         context = state[0][-1]
14         context = np.broadcast_to(context, (X.shape[0],
               context.shape[0], context.shape[1]))
15         X_and_context = np.concatenate((X, context), 2)
16         output, state = self.rnn(X_and_context, state)
17         output = self.dense(output).swapaxes(0, 1)
18         return output, state
```

## Paddle

```
1  class Seq2SeqDecoder(d2l.Decoder):
2      def __init__(self, vocab_size, embed_size,
           num_hiddens, num_layers, dropout=0, **kwargs):
3          super(Seq2SeqDecoder, self).__init__(**kwargs)
4          self.embedding = nn.Embedding(vocab_size,
               embed_size)
5          weight_attr = paddle.ParamAttr(initializer=nn.
               initializer.XavierUniform())
6          weight_ih_attr = paddle.ParamAttr(initializer=nn.
               initializer.XavierUniform())
7          weight_hh_attr = paddle.ParamAttr(initializer=nn.
               initializer.XavierUniform())
8          self.rnn = nn.GRU(embed_size + num_hiddens,
               num_hiddens, num_layers, dropout=dropout,
                   time_major=True, weight_ih_attr
                       =weight_ih_attr,
                       weight_hh_attr=
                       weight_hh_attr)
10         self.dense = nn.Linear(num_hiddens, vocab_size,
               weight_attr=weight_attr)
11
12     def init_state(self, enc_outputs, *args):
13         return enc_outputs[1]
14
15     def forward(self, X, state):
16         X = self.embedding(X).transpose([1, 0, 2])
17         context = state[-1].tile([X.shape[0], 1, 1])
18         X_and_context = paddle.concat((X, context), 2)
19         output, state = self.rnn(X_and_context, state)
20         output = self.dense(output).transpose([1, 0, 2])
21         return output, state
```

Figure 2: Implementing *RNN Decoder for Seq2Seq Learning* in PyTorch, TensorFlow, MXNet and Paddle. [Return to Section 3.2]

tion of the source code, then predict the output of the source code, and finally translate it, with the condition that the translated code must successfully execute.

The specific translation prompts are shown in Table 19.

| | Method | C | C++ | C# | Go | VB | Python | Java | PHP |
|---|---|---|---|---|---|---|---|---|---|
| | Naive | – | 14.61 | 9.08 | 4.52 | 1.64 | 2.17 | 11.23 | 2.09 |
| | OtO | – | 11.13±0.28 | 4.77±0.85 | 8.18±0.40 | 2.02±0.23 | 1.85±0.14 | 6.50±2.91 | 1.71±0.32 |
| C | OtM | – | 12.33±0.58 | 7.56±0.98 | 9.58±0.74 | 2.22±0.89 | 2.82±0.58 | 8.92±0.67 | 3.42±0.09 |
| | MtO | – | 7.99±1.94 | 5.35±0.41 | 6.94±0.83 | 3.77±0.44 | 2.09±0.25 | 8.65±0.52 | 1.93±0.49 |
| | MtM | – | 9.61±0.32 | 7.15±1.41 | 7.54±0.33 | 2.30±0.71 | 2.11±0.43 | 8.29±0.45 | 3.30±0.92 |
| | Naive | 14.88 | – | 10.08 | 3.87 | 3.76 | 1.69 | 11.39 | 1.92 |
| | OtO | 10.58±0.37 | – | 6.52±1.96 | 6.77±1.08 | 2.34±0.90 | 1.82±0.37 | 9.41±1.59 | 1.50±0.03 |
| C++ | OtM | 13.32±2.46 | – | 9.87±1.15 | 9.83±1.37 | 2.62±1.15 | 2.93±0.18 | 12.10±1.51 | 2.87±0.82 |
| | MtO | 10.54±2.33 | – | 6.98±1.46 | 7.29±0.73 | 3.57±0.74 | 2.28±0.18 | 7.82±1.26 | 1.96±0.20 |
| | MtM | 9.92±1.14 | – | 7.79±1.49 | 7.70±0.48 | 2.06±0.74 | 2.38±0.30 | 10.20±1.20 | 4.12±0.81 |
| | Naive | 9.05 | 10.03 | – | 5.05 | 7.20 | 1.83 | 13.60 | 2.58 |
| | OtO | 5.41±1.25 | 6.45±1.68 | – | 7.10±0.63 | 9.42±5.50 | 1.73±0.46 | 10.55±1.67 | 1.97±0.76 |
| C# | OtM | 7.42±1.82 | 8.99±0.17 | – | 9.81±0.68 | 6.13±2.92 | 2.80±0.43 | 11.92±2.52 | 4.55±0.81 |
| | MtO | 6.37±0.69 | 6.13±1.60 | – | 7.53±0.94 | 9.92±1.00 | 2.47±0.62 | 10.16±1.40 | 1.70±0.20 |
| | MtM | 6.19±0.88 | 7.22±0.93 | – | 9.36±0.92 | 4.60±2.87 | 2.23±0.36 | 12.50±1.35 | 4.70±0.27 |
| | Naive | 4.52 | 3.75 | 5.04 | – | 2.46 | 3.00 | 6.56 | 2.14 |
| | OtO | 5.65±0.41 | 6.88±0.47 | 4.70±1.28 | – | 1.87±0.54 | 2.89±0.16 | 5.41±1.06 | 2.48±0.23 |
| Go | OtM | 6.06±0.99 | 6.56±0.89 | 7.28±0.82 | – | 2.12±0.65 | 3.32±0.36 | 8.79±1.06 | 3.48±0.20 |
| | MtO | 5.41±0.63 | 4.77±0.76 | 6.89±0.33 | – | 2.94±0.63 | 3.12±0.48 | 7.81±0.32 | 1.79±0.51 |
| | MtM | 5.18±1.23 | 6.06±0.28 | 7.12±0.72 | – | 1.82±0.93 | 2.83±0.50 | 8.99±1.28 | 2.55±0.40 |
| | Naive | 1.64 | 3.76 | 7.29 | 2.54 | – | 1.42 | 2.89 | 0.46 |
| | OtO | 3.96±0.20 | 5.25±0.68 | 17.34±1.79 | 5.85±0.64 | – | 1.34±0.06 | 5.51±0.43 | 0.61±0.18 |
| VB | OtM | 5.15±0.15 | 6.10±0.30 | 19.63±1.03 | 7.83±0.45 | – | 2.02±0.32 | 7.91±0.92 | 2.02±0.20 |
| | MtO | 4.25±0.86 | 4.15±1.09 | 11.96±1.96 | 6.38±0.45 | – | 1.66±0.58 | 7.26±1.39 | 1.18±0.33 |
| | MtM | 5.18±0.65 | 5.09±0.15 | 14.13±1.83 | 6.97±1.30 | – | 2.34±0.11 | 8.39±0.77 | 2.92±0.22 |
| | Naive | 1.73 | 1.14 | 1.44 | 2.50 | 0.89 | – | 1.82 | 1.28 |
| | OtO | 4.35±0.44 | 3.66±0.95 | 5.38±0.76 | 5.67±0.34 | 2.53±1.11 | – | 4.70±0.07 | 3.24±0.60 |
| Python | OtM | 4.51±1.33 | 5.18±0.52 | 6.48±0.71 | 6.04±0.96 | 1.50±0.56 | – | 6.29±0.39 | 6.55±0.94 |
| | MtO | 4.70±0.28 | 3.44±0.76 | 5.48±0.55 | 4.80±0.63 | 2.42±0.18 | – | 5.32±0.66 | 2.74±0.39 |
| | MtM | 4.54±0.67 | 4.74±0.14 | 5.69±0.72 | 5.84±0.36 | 2.09±0.94 | – | 6.45±0.11 | 4.49±0.63 |
| | Naive | 11.23 | 11.15 | 13.44 | 6.57 | 2.84 | 2.24 | – | 2.60 |
| | OtO | 7.15±1.59 | 9.09±1.00 | 10.92±1.61 | 11.05±0.52 | 2.66±0.13 | 2.54±0.35 | – | 2.05±0.95 |
| Java | OtM | 9.27±1.52 | 9.33±0.86 | 13.52±0.95 | 12.57±0.96 | 2.92±0.78 | 3.98±0.86 | – | 6.20±1.08 |
| | MtO | 6.69±0.83 | 6.32±1.66 | 10.57±1.16 | 8.79±0.44 | 3.91±0.57 | 2.86±0.36 | – | 2.48±0.34 |
| | MtM | 6.56±0.33 | 7.60±0.46 | 9.40±1.08 | 8.30±0.87 | 2.20±0.85 | 2.53±0.15 | – | 3.57±0.94 |
| | Naive | 1.79 | 1.51 | 2.36 | 2.00 | 0.37 | 1.30 | 2.30 | – |
| | OtO | 4.18±1.44 | 2.91±0.43 | 6.45±0.43 | 5.85±0.54 | 0.42±0.09 | 1.98±0.26 | 6.47±0.59 | – |
| PHP | OtM | 3.78±1.80 | 1.93±1.35 | 5.51±0.60 | 5.13±0.60 | 0.56±0.36 | 2.70±0.80 | 5.51±1.91 | – |
| | MtO | 4.97±1.09 | 3.69±0.46 | 6.08±3.03 | 6.44±0.78 | 2.28±0.12 | 3.55±0.18 | 8.66±0.84 | – |
| | MtM | 6.03±0.61 | 5.15±0.67 | 10.64±1.27 | 6.69±0.31 | 1.69±0.57 | 2.46±0.46 | 7.87±1.90 | – |

Table 14: BLEU scores from different multilingual modeling strategies by fine-tuning the pre-trained CodeT5+_220M model (220M is the model size) (Wang et al., 2023b). Naive denotes Naive Copy, which directly duplicates the source code as the target code without making any modifications. Method OtO, OtM, MtO, and MtM denote One-to-One, One-to-Many, Many-to-One, and Many-to-Many, respectively. The rows correspond to the source language while the columns correspond to the target language. We run each experiment with three different random seeds and report the mean and standard deviation of BLEU scores. [Return to Section 4.1.]

| Method | Translation Prompt |
|---|---|
| **Role Assignment #1** | `"role": "system", "content": "Your are a code translation system.", "role": "user", "content": "Please provide the [TL] translation for the following [SL] code:[SC]` |
| **Role Assignment #2** | `"role": "system", "content": "You are a code translation system that specializes in [SL] and [TL] programming languages.", "role": "user", "content": "Please provide the [TL] translation for the following [SL] code:[SC]` |
| **Role Assignment #3** | `"role": "system", "content": "You are a programmer proficient in multiple programming languages.", "role": "user", "content": "Please provide the [TL] translation for the following [SL] code:[SC]` |
| **Role Assignment #4** | `"role": "system", "content": "You are a programmer proficient in [SL] and [TL] programming languages.", "role": "user", "content": "Please provide the [TL] translation for the following [SL] code:[SC]` |
| **Polite Inquiry #1** | `Please translate the following [SL] code into [TL] code:[SC]` |
| **Polite Inquiry #2** | `Can you rewrite this [SL] code in [TL]?[SC]` |
| **Clarify Usage** | `Translating [SL] to [TL] ensures that Python code can be executed.[SC]` |
| **Divide & Conquer** | `Translate [SL] to [TL]:[SC]` |

Table 15: Translation prompts for prompt variants and contextual strategies on ChatGPT. [SL] refers to the source language, [SC] refers to the source code, [TL] refers to the target language. [Return to Section 4.3]

| Strategy | Temp. | EM | BLEU | CodeBLEU | DSR@0 | Strategy | Top-K | EM | BLEU | CodeBLEU | DSR@0 |
|---|---|---|---|---|---|---|---|---|---|---|---|
| | 0 | 0.57 | **10.83** | 24.45 | 48.57% | | 0 | 0.57 | **10.83** | **24.45** | 48.57% |
| | 0.2 | 0.57 | 10.82 | 24.43 | 47.71% | | 0.2 | 0.57 | 10.81 | **24.45** | 48.29% |
| Top-K = 0 | 0.4 | 0.57 | 10.80 | 24.37 | 48.00% | Temp. = 0 | 0.4 | 0.57 | 10.81 | 24.44 | 48.29% |
| | 0.6 | 0.57 | 10.78 | 24.45 | 48.00% | | 0.6 | 0.57 | 10.82 | 24.42 | 48.29% |
| | 0.8 | 0.57 | 10.82 | **24.49** | 47.71% | | 0.8 | 0.57 | 10.80 | 24.38 | 47.71% |
| | 1.0 | 0.57 | 10.81 | 24.41 | 48.00% | | 1.0 | 0.57 | 10.81 | 24.38 | 48.00% |

Table 16: Code translation performance of ChatGPT under different parameter settings. *Temp.* refers to temperature. [Return to Section 4.3]

**Self-debug@0**

Translate [source_language] to [target_language]: [source_code]. `CTO`

🟢 Here is the [target_language] code equivalent
of the given [source_language] code: [translated_code].

- - - - - - - - - - - - - - - - - - - - - - - - - - - - - - - - - -

**Self-debug@n**

The above python code executes with the following errors,
please correct them. [Compiler reports errors] `CTO`

🟢 Here is the modified [target_language] code: [translated_code].

Table 17: A simple demo: Translation prompting of ChatGPT in the multi-round debugging strategy. The content in red is returned by the compiler. [Return to section 4.3.]

**Zero-shot prompting**

Does the following Python code execute? [python_code]. `CTO`

🟢 Yes, the Python code executes without errors.

Please predict the executed output of the Python code above. `CTO`

🟢 The predicted execution result of the Python code above is [output].

**Few-shot prompting**

This is a executable Python code [python_code], and this is a Python code [python_code]
that cannot be executed. Does the following Python code execute? [python_code]. `CTO`

🟢 Yes, the Python code executes without errors.

Please predict the executed output of the Python code above. `CTO`

🟢 The predicted execution result of the Python code above is [output].

Table 18: Two simple demos: prompting in fuzzy execution experiments. [Return to Section 6.]

| Method | Translation Prompt |
|---|---|
| **One-Shot** | `Here is an example of a`
`translation from [ESL] to`
`[ETL].[ESL]: [ESC], [ETL]: [ETC].`
`Please imitate this example to`
`translate following code from [SL]`
`to [TL]:[TC].` |
| **One-Shot #1** | `where [ESL]≠[SL] and [ETL]=[TL]` |
| **One-Shot #2** | `where [ESL]=[SL] and [ETL]=[TL]` |
| **One-Shot #3** | `where [ESL]≠[SL] and [ETL]≠[TL]` |
| **CoT #1** | `1.Please explain the function of`
`the following [SL] code, which`
`is limited to 200 words.[SC]`
`2.Please translate into [TL]`
`code according to the following`
`[SL] code and its functional`
`description.[SL]:[SC].Function`
`description:[DSC]` |
| **CoT #2** | `First, understand the function of`
`the following [SL] code.  Then,`
`translate the [SL] code into [TL]`
`code while keeping the function`
`unchanged.[SC]` |
| **CoT #3** | `First, understand the`
`functionality of the following`
`[SL] code and predict the`
`execution output.  Then, translate`
`the [SL] code into [TL] while`
`maintaining the same functionality,`
`ensuring that the translated code`
`can be successfully executed.[SL]` |
| **CoT #4** | `First, learn how to translate`
`[ESL] code to [ETL] based on the`
`example, [SL]:[ESC],[TL]:[ETC].`
`Then, understand the functionality`
`of the following [SL] code and`
`predict the execution output,`
`[SL]:[SC]. Finally, translate`
`the [SL] code into [TL] while`
`maintaining the same functionality,`
`ensuring that the translated code`
`can be successfully executed.` |

Table 19: ChatGPT translation prompts on one-shot & CoT strategies. [SL] refers to the source language, [SC] refers to the source code, [TL] refers to the target language,[ESL] refers to the source language in the example, [ESC] refers to the source code in the example, [ETL] refers to the target language in the example, [ETC] refers to the target code in the example, [DSC] refers to the natural language description of the source code. [Return to Section 4.3]