# OpenReview forum: "CodeTransOcean: A Comprehensive Multilingual Benchmark for Code Translation"
_EMNLP/2023/Conference — EMNLP 2023 Findings_

### Official Review · Reviewer_ebpA · 2023-08-04

**Soundness:** 3

**Excitement:**

3: Ambivalent: It has merits (e.g., it reports state-of-the-art results, the idea is nice), but there are key weaknesses (e.g., it describes incremental work), and it can significantly benefit from another round of revision. However, I won't object to accepting it if my co-reviewers champion it.

**Paper Topic And Main Contributions:**

This paper introduces a comprehensive multilingual benchmark for code translation tasks. The benchmark consists of 4 new datasets including MultilingualTrans for  translations between multiple popular PLs, NicheTrans for translating between niche PLs and popular ones, LLMTrans for evaluating compilability of translated code by LLMs, and DLTrans for translating deep learning code across different frameworks. This paper also proposes a new evaluation metric Debugging Success Rate@K for program-level code translation and evaluates both open LLMs such as CodeT5+ and closed-source models such as ChatGPT on these datasets. Given its large scale and comprehensiveness, this benchmark would be beneficial for measuring models’ code translation capabilities. A minor downside of this paper is that lots of results are placed into the Appendix, which violates the self-contained requirement of the main content.

**Questions For The Authors:**

* Do you analyze whether there are overlap between CodeTransOcean and the prior code translation datasets?
* Can you elaborate how can CodeTransOcean complement to prior code translation datasets?

**Reasons To Accept:**

* This is a novel and comprehensive code translation benchmark, which is beneficial for measuring LLMs’ code translation capabilities and contributes to the development of code LLMs.
* This paper provides comprehensive evaluation of both open LLMs such as CodeT5+ and closed-source models such as ChatGPT on these datasets. Besides, it proposes a new evaluation metric Debugging Success Rate@K for program-level code translation.

**Reasons To Reject:**

* The presentation of the paper can be improved, some of the important experimental results are placed in the  Appendix, which violates the self-contained requirement of the main content.

**Reproducibility:**

4: Could mostly reproduce the results, but there may be some variation because of sample variance or minor variations in their interpretation of the protocol or method.

**Reviewer Confidence:**

4: Quite sure. I tried to check the important points carefully. It's unlikely, though conceivable, that I missed something that should affect my ratings.

**Typos Grammar Style And Presentation Improvements:**

The presentation of the paper can be improved, some of the important experimental results are placed in the  Appendix, which violates the self-contained requirement of the main content.

---

> ### Author Rebuttal · Authors · 2023-08-28
>
> We thank the reviewer for appreciating our contribution of (1) providing a novel and comprehensive code translation benchmark, which is beneficial for measuring code translation capabilities of LLMs and contributes to the development of code LLMs, (2) conducting comprehensive evaluation of both open-source LLMs such as CodeT5+ and closed-source models such as ChatGPT on these datasets, and (3) proposing a new evaluation metric Debugging Success Rate@K for program-level code translation. Thank you very much for your constructive feedback and valuable suggestions. Below we address all of your questions and concerns.
>
> **Q1: Some important experimental results are in Appendix.**
>
> **Response:**
> We structure the paper based on two considerations. Firstly, the primary contributions of this paper are introducing the new CodeTransOcean benchmark for code translation. Therefore, we allocated the majority of the main body of the paper to elaborate the motivation and the process for creating this benchmark and dataset analysis, to demonstrate the efficacy of multilingual modeling on code translation, to propose a new evaluation metric, and to report comprehensive evaluations and studies of the representative LLM ChatGPT on these datasets.
> Secondly, due to the page limit, we put some experimental details and some experimental result tables such as those of CodeT5+ (Table 10-13) in Appendix. However, **our paper is self-contained**, since for **all** these tables of experimental results that we put in Appendix, **we have included the text describing the purpose of the corresponding experiment and the detailed discussions corresponding to the results in the main body**. In the main body, for CodeT5+ results, Line 401-434 provides detailed analyses and discussions for Table 10, Line 435-447 for Table 11, Line 448-455 for Table 12 and 13, and Line 456-466 provides an overall analysis of the observations on Table 10-13. In the main body, for ChatGPT results, Line 569-580 provides detailed discussions for Table 15 of ChatGPT performance under different Top-K and Temperature parameters, and Line 592-618 details the experimental setup and results for Table 16 and Table 19, evaluating the effect of multi-round self-debugging on DSR.
> In the revised version, we will refine the organization of the content and move the key result tables from Appendix to the main body.
>
> **Q2: Whether there are overlap between CodeTransOcean and the prior code translation datasets**
>
> **Response:**
> We confirm that **there is no overlap between all the four datasets included in our CodeTransOcean and the existing code translation datasets**. Our work is not a simple data amplification over prior works. We provide three novel and unique datasets NicheTrans, DLTrans and LLMTrans for code translation tasks and our MultilingualTrans dataset is advantageous in larger size, more balanced data distribution across various programming languages, practicality of language pairs, and data quality compared to the existing code translation datasets (explained in lines 191-196, 259-264, 1200-1215 of the paper). For the construction of CodeTransOcean, we carefully selected data from [Rosetta Code](https://rosettacode.org/wiki/Rosetta_Code) and [Dive into Deep Learning](https://d2l.ai/) from multiple code sites. When building our datasets, we carefully compared with the data sources of other code translation datasets to ensure that **the same data sources are not used**.
> We will add the specific data sources for each dataset in Table 2 of our paper to further clarify this point.
>
> **Q3: How can CodeTransOcean complement to prior code translation datasets?**
>
> **Response:**
> We elaborate on the complementary relationship between CodeTransOcean and the existing code translation datasets as follows. Prior to our work, the most comprehensive code translation dataset, CoST, covers seven programming languages. The XTest paper claims their dataset covers nine programming languages, but has not been open-sourced. In contrast, we released the MultilingualTrans dataset, which encompasses 8 popular programming languages and is advantageous in larger size, more balanced data distribution across various programming languages, practicality of language pairs, and data quality compared to the existing code translation datasets (explained in lines 191-196, 259-264, 1200-1215 of the paper). Moreover, we have also introduced the NicheTrans dataset, which covers 37 programming languages with medium or low usage rates. In addition, we have specifically designed the LLMTrans dataset for evaluating LLMs on code translation and provided the DLTrans dataset that spans four major deep learning frameworks. It is worth noting that while the MultilingualTrans dataset shares some similarities with some previous datasets, to date MultilingualTrans covers the broadest range of programming languages and the largest number of unit tests and comprises **program-level** parallel data, while the data samples in previous datasets are usually simple parallel data at the function-level. The three datasets LLMTrans, DLTrans and NicheTrans are all created for the first time, are all novel and unique, and are also **program-level** parallel data. **Therefore, CodeTransOcean covers the largest number of programming languages to date and is a valuable resource in the field of code translation, as acknowledged by Reviewer RZux and Reviewer XoXb. More importantly, CodeTransOcean fits perfectly with EMNLP's interest in "[New data resources, particularly for low-resource languages](https://2023.emnlp.org/calls/main\_conference\_papers/\#contributions)".**
> Furthermore, we have provided evaluation tools and an automated pipeline for code translation using LLMs. To further help researchers exploit this benchmark, we have also established corresponding baselines on multiple strategies for each dataset of CodeTransOcean.
>
> **Reproducibility: 4**
>
> **Response:**
> To ensure the transparency and reproducibility of our experiments, we have already provided all the relevant code, datasets and experimental results in the supplementary materials. We will make all datasets of CodeTransOcean, code, experimental results, and model checkpoints publicly available, so that researchers can reproduce all our results. We also created a website and leaderboard for this benchmark. To demonstrate these to reviewers, we uploaded the dataset, code, experimental results, website, and model checkpoints in [anonymous Github](https://anonymous.4open.science/r/CTO-5263/Readme.md) as a way to demonstrate the reproducibility of our work. All in all, based on our rigorous internal testings, we are confident that our experimental results are easily reproducible.
> In addition, Line 382-383 points out that the details of multilingual modeling strategies are in Appendix A.5, where A.5 provides the details of the specific hyperparameter settings and environment used in all experiments in Table 9. The reason that the training and evaluation data has not been widely available is because the training/evaluation data is from our CodeTransOcean and we have not yet disclosed the datasets before the anonymous review period.

---

### Official Review · Reviewer_XoXb · 2023-08-04

**Soundness:** 4

**Excitement:**

3: Ambivalent: It has merits (e.g., it reports state-of-the-art results, the idea is nice), but there are key weaknesses (e.g., it describes incremental work), and it can significantly benefit from another round of revision. However, I won't object to accepting it if my co-reviewers champion it.

**Missing References:**

[1] Zhu, Ming, Aneesh Jain, Karthik Suresh, Roshan Ravindran, Sindhu Tipirneni and Chandan K. Reddy. “XLCoST: A Benchmark Dataset for Cross-lingual Code Intelligence.” ArXiv abs/2206.08474 (2022): n. pag.
[2] Rithy, Israt Jahan, Hasib Hossain Shakil, Niloy Mondal, Fatema Sultana and Faisal Muhammad Shah. “XTest: A Parallel Multilingual Corpus with Test Cases for Code Translation and Its Evaluation*.” 2022 25th International Conference on Computer and Information Technology (ICCIT) (2022): 623-628.

**Paper Topic And Main Contributions:**

The paper introduces a novel multilingual benchmark for code translation, named "CodeTransOcean". Recognizing that existing code translation datasets typically focus on just one pair of popular programming languages, the authors created this comprehensive benchmark to cater to a wide variety of languages, reflecting the diversity in real-world applications. CodeTransOcean contains three distinct datasets designed to support translations between multiple popular programming languages, niche languages, and also to assess the reliability of translated code by large language models. Moreover, it incorporates a dataset specifically for translating deep learning code across different frameworks. The authors also pioneered multilingual modeling approaches for code translation and proposed a new metric for evaluating code translation at the program level. The paper includes an evaluation of a large language model, ChatGPT, using this new benchmark, and provides baseline results and an analysis of challenges in code translation.

**Questions For The Authors:**

- How did you decide which models to evaluate using the CodeTransOcean benchmark? Why methods that are assessed in previous work are not included in this paper?
- Why existing work like [1][2] is not referenced and compared in this paper?

**Reasons To Accept:**

- Well structured and easy to follow
- The proposed benchmark includes creating three unique datasets to support translations between various popular and niche programming languages, and to evaluate the reliability of translated code. In addition, they develop a cross-framework dataset specifically for translating deep learning code across different frameworks.
- Experiments demonstrate the potential of multilingual modeling approaches to improve code translation quality for various language pairs and increase training efficiency.

**Reasons To Reject:**

- Limited model evaluation: The paper primarily focuses on the evaluation of ChatGPT, neglecting other significant models such as Transformer, CodeBERT, DOBF, and TransCoder, and open-source LLMs like LLaMa. This narrows the scope of the benchmark's applicability and restricts comprehensive comparative analysis.
- Missing reference: By not referencing or discussing key works such as [1][2], the authors may miss out on important comparative analyses.

**Reproducibility:**

4: Could mostly reproduce the results, but there may be some variation because of sample variance or minor variations in their interpretation of the protocol or method.

**Reviewer Confidence:**

3: Pretty sure, but there's a chance I missed something. Although I have a good feel for this area in general, I did not carefully check the paper's details, e.g., the math, experimental design, or novelty.

**Typos Grammar Style And Presentation Improvements:**

In the paper, it would be beneficial to include the main results table in the main body of the text rather than in the appendix. This change would enable readers to easily relate the experimental results to the methods and discussions as they progress through the paper.

---

> ### Author Rebuttal · Authors · 2023-08-28
>
> We thank the reviewer for appreciating our contributions of creating unique datasets for code translation tasks, evaluating reliability of translated code, and providing empirical validations of potentials of multilingual modeling for improving translation quality and increasing training efficiency. We thank the reviewer very much for all thoughtful reviews and valuable feedback. Below are our responses to all of your questions and concerns.
>
> **Q1: Limited model evaluation.**
>
> **Response:** We respectfully disagree the reviewer's comment that “The paper primarily focuses on the evaluation of ChatGPT, neglecting other significant models such as Transformer, CodeBERT, DOBF, and TransCoder, and open-source LLMs like LLaMa.”
> In this work, we aim at investigating various multilingual training strategies and studying the potential of LLMs on code translation tasks, hence we did not report experimental results on a wide range of models; instead, we comprehensively evaluate the strategies and potentials using the best performing models. **We conducted careful model selections based on both prior works and our experimental validations**.
> Firstly, Table 2, 3, 4, 5 and Figure 6 in the CodeT5+ paper (Wang et al., 2023b) which is cited in our paper shows extensive performance evaluations of open-source and closed-source LLMs on code tasks.  The results show that CodeT5+ significantly outperforms prior models including encoder-only models such as RoBERTa, CodeBERT, and GraphCodeBERT, decoder-only models such as GPT-2 and CodeGPT, and encoder-decoder models such as CodeT5 and PLBART. Details of these baselines that CodeT5+ compared to are in Section 5 of the CodeT5+ paper.
> The results in the CodeT5+ paper show that CodeT5+ is the best performing **open-source pre-trained language model for code tasks**, and CodeT5+ outperforms LLaMA on code tasks. Since the sizes of the three datasets of MultilingualTrans, NicheTrans and DLTrans require fine-tuning, we chose the pre-trained CodeT5+ as the backbone for all multilingual training experiments, shown in Table 10-13. Secondly, we also evaluated CodeBERT, CodeT5 and CodeT5+ on MultilingualTrans, NicheTrans and DLTrans and observed that CodeT5+ significantly outperforms CodeBERT and CodeT5.
>
> For evaluations on our LLMTrans dataset, we chose GPT-3.5 to evaluate because GPT-4 and GPT-3.5 are the Top-2 LLMs on code tasks, as shown in Table 2 of the CodeT5+ paper and at the time of this work, we did not have access to GPT-4 API. Hence GPT-3.5 is the best performing LLM available to us. Since GPT-3.5 has significantly better code task performance than LLaMA, as shown in Table 2 of the CodeT5+ paper, we focus on conducting comprehensive evaluations of various algorithms and studying potential of ChatGPT on code translation tasks, with results reported in Table 3, 4, and 19. In our revised version, we will add experimental results for CodeBERT, CodeT5, and LLaMA for the sake of completeness.
>
> Regarding DOBF and TransCoder that you mentioned, DOBF is a pre-training method related to programming languages, not a specific model for code tasks, hence not related to our work. TransCoder is a fully unsupervised neural transcompiler based on unsupervised machine translation approaches. Its code translation performance is mediocre and is not widely used or evaluated. Hence, we did not discuss this model among code translation models in Related Work nor compare to this model. We only cited the dataset used in the Transcoder work in Line 189-193 of our paper. We will add TransCoder into related code translation models in Related Work.
>
>
> **Q2: Missing reference [1][2].**
>
> **Response:** Thank you for suggesting these two papers. Regarding paper [1], the code intelligence benchmark it introduced comprises five sub-tasks, and its “Code Translation” section is actually based on augmenting the data from the CoST paper we already cited in Line 191-193. It is worth noting that both studies are authored by the same team, and their data was directly collected from the GeeksForGeeks website. From the perspective of contributions to code translation, there is no significant difference between these two datasets. Hence we did not cite paper [1]. However, to ensure completeness, we will add a reference to paper [1] in the revised version.
>
> As for paper [2], its main contribution is to propose a code translation dataset XTest containing nine programming languages with unit tests, but **it is not open-sourced**. Its contribution is similar to that of our MultilingualTrans dataset, which aims to expand code translation pairs between popular programming languages. As can be seen from Table 2 and Line 178-212 of our paper, we performed statistical analysis and verification of each dataset. However, the actual usability of the dataset in paper [2] cannot be analyzed or verified. We tried to contact the authors of paper [2] but there was no response. Hence we chose not to cite paper [2] in our paper.  Once we can access and validate this dataset, we will add comparisons and in-depth analysis of this paper in the revised version.
>
> As shown in the above analyses, it is clear that while paper [1] and [2] contribute to the field, **their works do not impact nor overshadow the contributions of our work**.
>
> **Q3: How did you decide which models to evaluate using CodeTransOcean? Why some models assessed in previous works are not included in this paper?**
>
> **Response:** Please refer to our response to Q1.
>
> **Q4: Why existing work like [1][2] is not referenced and compared in this paper?**
>
> **Response:** Please refer to our response to Q2.
>
> **Presentation Improvements: Would be beneficial to include the main results table in the main body of the text rather than in the appendix.**
>
> **Response:**
> We structure the paper based on two considerations. Firstly, the primary contributions of this paper are introducing the new CodeTransOcean benchmark for code translation. Therefore, we allocated the majority of the main body of the paper to elaborate the motivation and the process for creating this benchmark and dataset analysis, to demonstrate the efficacy of multilingual modeling on code translation, to propose a new evaluation metric, and to report comprehensive evaluations and studies of the representative LLM ChatGPT on these datasets.
> Secondly, due to the page limit, we only put some experimental details and some experimental result tables such as those of CodeT5+ (Table 10-13) in Appendix. However, **our paper is self-contained**, since for **all** these tables of experimental results that we put in Appendix, **we have included the text describing the purpose of the corresponding experiment and the detailed discussions corresponding to the results in the main body**. In the main body, for CodeT5+ results, Line 401-434 provides detailed analyses and discussions for Table 10, Line 435-447 for Table 11, Line 448-455 for Table 12 and 13, and Line 456-466 provides an overall analysis of the observations in Table 10-13. In the main body, for ChatGPT results, Line 569-580 provides detailed discussions for Table 15, and Line 592-618 details the experimental setup and results for Table 16 and 19. In the revised version, we will refine the organization of the content and move the key result tables from Appendix to the main body.
>
>
> **Reproducibility: 4**
>
> **Response:**
> To ensure the transparency and reproducibility of our experiments, we have already provided all the relevant code, datasets and experimental results in the supplementary materials. We will make CodeTransOcean's datasets, code, experimental results, and model checkpoints publicly available, so that researchers can reproduce all our results. We also created a website and leaderboard for this benchmark. To demonstrate these to reviewers, we uploaded the dataset, code, experimental results, website, and model checkpoints in [anonymous Github](https://anonymous.4open.science/r/CTO-5263/Readme.md) as a way to demonstrate the reproducibility of our work. All in all, based on our rigorous internal testings, we are confident that our experimental results are easily reproducible. In addition, Line 382-383 points out that the details of multilingual modeling strategies are in Appendix A.5, where A.5 provides the details of the specific hyperparameter settings and environment used in all experiments in Table 9. The reason that the training and evaluation data has not been widely available is because the training/evaluation data is from our CodeTransOcean and we have not yet disclosed the datasets before the anonymous review period.
>
> CodeT5+ (Wang et al., 2023b) Yue Wang, Hung Le, Akhilesh Deepak Gotmare, Nghi D. Q. Bui, Junnan Li, and Steven C. H. Hoi. 2023b. CodeT5+: Open code large language models for code understanding and generation. arxiv:2305.07922.

---

### Official Review · Reviewer_RZux · 2023-08-10

**Soundness:** 3

**Excitement:**

3: Ambivalent: It has merits (e.g., it reports state-of-the-art results, the idea is nice), but there are key weaknesses (e.g., it describes incremental work), and it can significantly benefit from another round of revision. However, I won't object to accepting it if my co-reviewers champion it.

**Paper Topic And Main Contributions:**

This paper is about advancing research on code translation by constructing a large-scale comprehensive benchmark called CodeTransOcean. It addresses the problem of limited code translation datasets that usually focus on a single pair of popular programming languages. To tackle this issue and meet diverse requirements of real-world applications, the authors propose three novel multilingual datasets: MultilingualTrans, NicheTrans, and LLMTrans. Additionally, they include a cross-framework dataset called DLTrans for translating deep learning code across different frameworks.

The main contributions of this paper are:
1.	The creation of CodeTransOcean, a benchmark that supports the largest variety of languages for code translation.
2.	The development of multilingual modeling approaches for code translation, which improve the translation quality of both low-resource and high-resource language pairs and boost training efficiency.
3.	The proposal of a novel evaluation metric called Debugging Success Rate@K for program-level code translation.
4.	The evaluation of LLM ChatGPT on their datasets and investigation of its potential for fuzzy compilation predictions.
5.	The establishment of baselines for CodeTransOcean and analysis of challenges of code translation to guide future research.

These contributions align with the EMNLP's interests in new data resources, approaches for data and compute efficiency, and publicly available software and pre-trained models.


**Reasons To Accept:**

1. Since the emergence of ultra-large language models, the focus of the NLP community has increasingly shifted from model-driven to data-driven research. This paper contributes valuable new paired data for code translation tasks, making it a noteworthy and commendable effort.
2. The NicheTrans Dataset and LLMTrans Dataset, which involve more specialized fields, hold significant value. In the context of industrial-level code, the framework migration for these less popular fields urgently requires high-quality annotated data. Such data has the potential to facilitate the execution of related projects and inspire subsequent research within the community.
3. The author engages in a thorough performance analysis of the current code translation task on large language models, as well as in-depth ablation experiments. These efforts are commendable and deserving of recognition.


**Reasons To Reject:**

1. Regarding the evaluation metrics for code translation tasks, earlier metrics such as CodeBLEU had shortcomings in reflecting the true effectiveness of the generated code. Subsequently, the community introduced metrics like executable accuracy and computation accuracy. However, these methods were constrained by the cost of constructing the execution environment, resulting in limited versatility. The new metric proposed by the author still relies on the compilation environment, and thus, it does not fully address the core issue at hand.

2. Besides the scale of the collected data, the quality of the gathered code pairs is also crucial for current code translation task datasets. Some recent new datasets, such as Avatar and TransCoder, often include unit tests and executable environments. However, most of the author's experimental section only involves metrics like EM, which do not provide a clear explanation of the code pair quality.


**Reproducibility:**

3: Could reproduce the results with some difficulty. The settings of parameters are underspecified or subjectively determined; the training/evaluation data are not widely available.

**Reviewer Confidence:**

5: Positive that my evaluation is correct. I read the paper very carefully and I am very familiar with related work.

---

> ### Author Rebuttal · Authors · 2023-08-28
>
> We thank the reviewer for appreciating our contributions of providing valuable new paired data for code translation tasks, facilitating execution of important specialized fields in real-world code translation applications (such as NicheTrans) and inspiring subsequent research, and our thorough performance analysis of LLMs on code translation tasks and in-depth ablation analysis. We sincerely thank the reviewer for all valuable feedback. Below we address all your questions and concerns.
>
> **Q1: Earlier match-based and execution-based metrics have shortcomings. The new metric proposed does not fully address the core issue at hand.**
>
> **Response:** Firstly, we would like to emphasize that the primary contribution of this paper is to establish a comprehensive code translation benchmark including new datasets for high-resource and low-resource programming languages, cross-deep-learning framework datasets, and for effectively evaluating LLMs on code translation tasks. Our goal is not to fully address all the issues in existing code translation evaluation metrics, such as neglecting executability and functional equivalence with match-based metrics and constrained by execution environment with execution-based metrics. We propose the Debugging Success Rate@K (DSR@K) metric to meet the needs of **code debugging** with LLMs and to complement the previously proposed Pass@k metric (Line 1143-1146). As explained in Line 484-492, DSR@K is the first metric designed to accurately reflect real-world software development scenarios. With the promising capabilities of LLMs for code debugging, we consider this new metric both practical and reasonable.
> Note that we did point out the limitations of our current automatic pipeline AutoTransCompiler and the DSR@K metric in Sections
> Limitations and Related Work (Line 696-734 and 1101-1163), as AutoTransCompiler currently only supports Python as the target
> language hence there is no constraint from compilation/execution environment.
>
> We consider two potential solutions to address core issues of code translation evaluation metrics. **One solution is Extending AutoTransCompiler to support a wide range of programming languages** by implementing automatic detection of code dependencies and automatic compilation and execution in a sandbox for safe execution. However, even recent commercial products such as [coderpad.io](https://rosettacode.org/wiki/Rosetta_Code) only provide a multilingual execution engine and cannot yet fully automate the entire process. There are still many technical challenges to overcome. **Another solution is further exploring our proposed fuzzy compilation**. As discussed in Section Limitations Line 731-779, we believe that fuzzy compilation using LLMs has the potential to address the limitations of current match-based and execution-based evaluation metrics. Although ChatGPT for fuzzy compilation is not yet practical, we are exploring fine-tuning methods to improve it.
>
> **Q2: Need to provide a clear explanation of the code pair quality.**
>
> **Response:**
> Here we clarify how we ensured high data quality for our CodeTransOcean datasets.
>
> **(1) Data quality:**
> Section 3 Line 220-222 points out that the details of data quality control are in Appendix A.2, where A2.2 and A2.3 Line 1165-1215 elaborate how we ensure the quality of the data. We verify that for the MultilingualTrans dataset with compilation requirements, the compilability rate exceeds 90\%. We verify that the code pairs in each dataset of CodeTransOcean are functionally identical and confirm that CodeTransOcean is of high quality.
>
> **(2) Unit tests:** Our CodeTransOcean provides unit tests. The vast majority of the samples in our datasets provides explicit input and output, which is equivalent to unit tests. These input and output information is collected from our data source [rosettacode.org](https://rosettacode.org/wiki/Rosetta_Code) as the vast majority of tasks in the source provides a clear input and output. Our CodeTransOcean, with **over 200,000 unit tests** and **45 covered languages**, far exceeds the scales of Avatar and TransCoder  (Avatar includes Java and Python, with a total of 250 unit tests; TransCoder encompasses Java, C++, and Python with 2,112 unit tests).
>
> **(3) Execution environment:** As in our response to Q1, the focus of this paper is not to fully address all the issues in existing code translation evaluation metrics. In this paper, we provide an execution environment for Python, given that many code intelligence tasks, such as HumanEval, MBPP, and DS 1000, are based on Python. And our AutoTransCompiler currently supports automatically using LLMs to conduct code translation, compilation, debugging and calculating success rate, with the target language as Python.
>
> **(4) Evaluation metrics:** We respectfully disagree on the reviewer's comment that “most of the author’s experimental section
> only involves metrics like EM”. Except for our proposed **DSR@K** metric used in the experiments on the LLMTrans dataset, we used both **EM** and **BLEU** evaluation metrics in all other experiments (Table 3, Table 10-13, Table 15) following prior works. Table 3, 12, 15 also report CodeBLEU in addition to EM and BLEU.  We explained these metrics in detail in Appendix A.1.2. EM and BLEU have been extensively used in prior studies such as CodeTrans (NeurIPS 2021), CoST (AAAI 2022), Avatar (ACL 2023) and TransCoder (NeurIPS 2020). As explained in Sections Limitations Line 696-709 and Related Work Line 1101-1163, there are no other match-based metrics other than EM and BLEU that can support the 45 programming languages covered by our CodeTransOcean. Moreover, as described in our response in Q1, it would require great effort to implement execution-based metrics for this large number of programming languages.
>
> **(5) Version differences in the Avatar paper:** It is worth noting that when we wrote this paper, the Avatar version (Ahmad et al., 2021b) ([version 1](https://arxiv.org/abs/2108.11590v1)) that we cited was published on August 26, 2021, which did not include unit tests and an execution environment. The reviewer referred to the [version 2](https://arxiv.org/abs/2108.11590v2) published on May 4, 2023, which incorporated unit tests. According to the EMNLP 2023 guideline that “papers (whether refereed or not) appearing less than 3 months before the submission deadline are considered contemporaneous to your submission, and you are therefore not obliged to make detailed comparisons that require additional experimentation and/or in-depth analysis.”,  our work is contemporaneous to version 2 and does not need to compare to version 2.
>
> **Reproducibility: 3**
>
> **Response:**
> To ensure the transparency and reproducibility of our experiments, we have already provided all the relevant code, datasets and experimental results in the supplementary materials. We will make CodeTransOcean's datasets, code, experimental results, and model checkpoints publicly available, so that researchers can reproduce all our results. We also created a website and leaderboard for this benchmark. To demonstrate these to reviewers, we uploaded the dataset, code, experimental results, website, and model checkpoints in [anonymous Github](https://anonymous.4open.science/r/CTO-5263/Readme.md) as a way to demonstrate the reproducibility of our work. All in all, based on our rigorous internal testings, we are confident that our experimental results are easily reproducible.
> In addition, Line 382-383 points out that the details of multilingual modeling strategies are in Appendix A.5, where A.5 provides the details of the specific hyperparameter settings and environment used in all experiments in Table 9. The reason that the training and evaluation data has not been widely available is because the training/evaluation data is from our CodeTransOcean and we have not yet disclosed the datasets before the anonymous review period.

---

### Meta-Review · Area_Chair_nGQ3 · 2023-09-18

**Recommendation:** 4

**Metareview:**

This paper introduces comprehensive new data sets for code translation between many different programming languages.

Pros:
- Valuable new data sets, high number of programming languages covered. Much larger than other data sets in the space.
- Detailed experiments involving LLMs and code translation using these new data sets.
- Inclusion of even "niche" programming languages like Fortran, for which code translation can be useful as part of modernization of legacy systems.
- Reproducibility and extensibility: Datasets, code, experimental results, and model checkpoints are openly available, enabling further research down the line.
- Well structured and easy to follow.

Cons:
- Some concerns about missing comparisons with other data sets, and unclarity around the selection of the models that were chosen for initially evaluating the benchmark on. This can be addressed in a minor revision.

The main focus of this paper is the creation of the new data sets, not the experimental results on various models, so it's not a problem that the tables with the detailed experimental results only appear in the appendix, given that the top-level analysis of these experiments appears in the main body of the text.

---

### Decision · Program_Chairs · 2023-10-07

**Decision:**

Accept-Findings

**Comment:**

This paper introduces comprehensive new data sets for code translation between many different programming languages.

Pros:
- Valuable new data sets, high number of programming languages covered. Much larger than other data sets in the space.
- Detailed experiments involving LLMs and code translation using these new data sets.
- Inclusion of even "niche" programming languages like Fortran, for which code translation can be useful as part of modernization of legacy systems.
- Reproducibility and extensibility: Datasets, code, experimental results, and model checkpoints are openly available, enabling further research down the line.
- Well structured and easy to follow.

Cons:
- Some concerns about missing comparisons with other data sets, and unclarity around the selection of the models that were chosen for initially evaluating the benchmark on. This can be addressed in a minor revision.

The main focus of this paper is the creation of the new data sets, not the experimental results on various models, so it's not a problem that the tables with the detailed experimental results only appear in the appendix, given that the top-level analysis of these experiments appears in the main body of the text.